# L4Q: Parameter Efficient Quantization-Aware Fine-Tuning on Large Language Models

## Abstract

Due to the high memory and computational costs associated with large language models (LLMs), model compression techniques such as quantization, which reduces inference costs, and parameter-efficient fine-tuning (PEFT) methods like Low-Rank Adaptation (LoRA), which reduce training costs, have gained significant popularity. This trend has spurred active research into quantization-aware PEFT techniques, aimed at maintaining model accuracy while minimizing memory overhead during both inference and training. Previous quantization-aware PEFT methods typically follow a two-step approach: first, applying post-training quantization (PTQ) to model weights, followed by PEFT on the quantized model. However, recovering from the quantization error introduced by PTQ through fine-tuning has proven challenging. Additionally, most PTQ-based PEFT methods result in a mixture of low-precision quantized weights and high-precision adapter weights, limiting the efficiency of full quantization during inference. While a previous method attempted to address these issues, it still suffers from limited adaptability due to the constrained LoRA parameter structure required to produce fully-quantized models. To overcome these challenges, we propose L4Q, a method that integrates Quantization-Aware Training (QAT) with LoRA to effectively reduce quantization error. By employing a memory-optimized layer design, L4Q significantly reduces QAT's memory overhead while producing fully-quantized weights, enabling effective adaptation to downstream tasks. Our experiments demonstrate that this combined approach to quantization and fine-tuning achieves superior accuracy compared to decoupled fine-tuning schemes, particularly in sub-4-bit quantization, positioning L4Q as an efficient QAT solution. Using the LLaMA model families and instructional datasets, we showcase L4Q's capabilities in language tasks and few-shot learning.

## 1 Introduction

Given their impressive scalability, Large Language Models (LLMs) such as GPT, OPT, PaLM, and LLaMA (Brown et al., 2020; Zhang et al., 2022; Chowdhery et al., 2024; Touvron et al., 2023a;b) are popular in natural language processing. However, their substantial memory and computational demands pose challenges for practical deployment, making model compression (Han et al., 2015) crucial for LLM deployment. Quantization is a prominent method that reduces model size by lowering the bit precision of model parameters (Hubara et al., 2017), so LLM quantization has been actively studied (Liu et al., 2024; Xiao et al., 2023; Frantar et al., 2023; Dettmers & Zettlemoyer, 2023). These quantization methods are generally divided into two categories: post-training quantization (PTQ) and quantization-aware training (QAT). QAT effectively reduces quantization error by integrating quantization into the training process, where both model weights and quantization parameters, are trained together (Esser et al., 2020; Bhalgat et al., 2020). However, applying QAT to LLM quantization is challenging due to the significant memory overhead it incurs during training. As a result, PTQ, which applies quantization without retraining the pre-trained model weights and with minimal calibration of the quantization parameters, is widely adopted for LLM quantization (Xiao et al., 2023; Lin et al., 2024; Heo et al., 2024).

Concurrently, to enhance the problem-solving abilities of LLMs for specific applications, fine-tuning pre-trained LLMs on downstream tasks is crucial as it improves accuracy on target tasks and related tasks (Wei et al., 2022; Scialom et al., 2022). However, fine-tuning is a resource-intensive pro-

cess due to the large number of model weight parameters involved. Parameter-efficient fine-tuning (PEFT) addresses this issue (Hu et al., 2022; Li & Liang, 2021; Liu et al., 2022a; Wang et al., 2023) by training a small subset of parameters while freezing the majority of pre-trained weights. One of the most prominent techniques within PEFT is Low-Rank Adaptation (LoRA) (Hu et al., 2022), which inserts trainable rank decomposition matrices into each layer to represent updates to the frozen weights.

The integration of quantization and PEFT holds significant potential for developing highly efficient and accurate LLMs for downstream tasks. Recent research has introduced quantization-aware PEFT approaches to achieve high-quality quantized models (Dettmers et al., 2024b; Kim et al., 2024; Xu et al., 2023; Li et al., 2024). Previous works involve a two-stage optimization strategy: first, a PTQ technique, such as GPTQ (Frantar et al., 2023), is applied to pre-trained LLMs for compression. Then, these quantized LLMs undergo PEFT, such as LoRA, where quantized weights are kept fixed and only the LoRA parameters are fine-tuned. While fine-tuning can mitigate the effects of quantization errors, separating quantization and fine-tuning into distinct state hin-

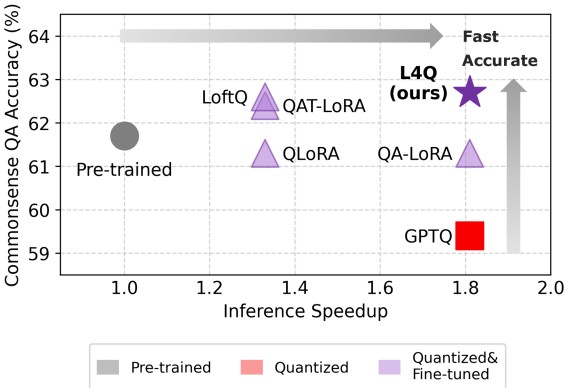

Figure 1: A diagram of model fine-tuning and quantization represented with an example of LLaMA-1 7B model. L4Q produces fast and accurate quantized model.

ders the models from achieving the best accuracy. Furthermore, as high-precision LoRA parameters are adopted alongside the quantized weight matrix, these methods eventually produce mixed-precision models, which limits the efficiency of full quantization during inference. Recently, QA-LoRA (Xu et al., 2023) addresses this issue by strictly constraining the LoRA parameter structure to integrate with quantization parameters, but this constraint can limit the fine-tuning capability.

In this paper, we propose a novel quantization-aware fine-tuning technique, named L4Q (Low-rank adaptive Learning quantization for LLMs). L4Q addresses the limitations of PTQ-based PEFT methods by introducing QAT alongside LoRA. While QAT and LoRA have advantages in reducing quantization error and enabling memory-efficient training, respectively, their straightforward integration diminishes the benefits of each approach. Therefore, L4Q carefully integrates these two approaches to properly leverage their advantages. First, L4Q applies the quantization process after fully combining the model weights and LoRA parameters in the linear layer. This approach produces a fully-quantized model that enables memory-efficient and fast inference without limiting the training capabilities of either QAT or LoRA. Moreover, to preserve the memory-efficient nature of LoRA during training, we design the backpropagation path of L4Q to eliminate the need to store weight gradients required for QAT. Finally, the full integration of QAT and LoRA in the proposed L4Q allows for the joint optimization of both the quantization and LoRA parameters, thereby improving of quantized LLMs. As a result, L4Q significantly improves the accuracy of quantized models while maintaining low memory costs during both inference and training, and achieves inference speed comparable to state-of-the-art approaches, making it a more effective solution compared to previous works, as illustrated in Figure 1.

## 2 BACKGROUNDS

### 2.1 PEFT WITH LoRA

LoRA inserts the rank-decomposition matrices composed of a pair of parameter matrices $A \in R^{r \times i}$ and $B \in R^{o \times r}$. Here, $i$ and $o$ represent the size of input and output dimensions of the original weight matrix, respectively. $r \ll i, o$ is the rank of the LoRA matrices, and $\alpha$ is a constant that adjusts the influence of the LoRA matrices. During the fine-tuning process, the pre-trained weight matrix $W_0 \in R^{o \times i}$ is frozen, preserving the pre-trained features. For a given input activation $X \in R^{i \times s \times b}$ ($s$: sequence length, $b$: batch size), the output $Y \in R^{o \times s \times b}$ of a layer utilizing LoRA

is computed as follows:

$$Y = (W_0 + \alpha BA)X = W_0 X + \alpha BAX \tag{1}$$

The fine-tuning of the LoRA parameters is guided by the gradient of a loss function $L$, which is calculated with respect to each parameter matrix. The gradients are derived as follows:

$$\frac{\partial L}{\partial A} = \alpha \frac{\partial L}{\partial \tilde{X}} X^\top, \quad \frac{\partial L}{\partial B} = \alpha \frac{\partial L}{\partial Y} \tilde{X}^\top \tag{2}$$

Here, $\tilde{X} := AX$ represents the intermediate input activation of $B$, which is the transformation of $X$ by $A$. These gradients guide the adjustment of the LoRA parameters to minimize the loss and more accurately approximate the necessary updates to the original model weights.

## 2.2 QUANTIZATION

Uniform quantization is a widely used quantization scheme due to its simplicity and broad compatibility with various computing kernels and hardware units (Liu et al., 2022b). Therefore, considering the adaptability on the broad quantization scheme, including both weight-only and activation quantization scenarios, we refer to the term 'quantization' specifically as uniform quantization throughout this paper. A common practice is to organize a quantization group consisting of a certain number of consecutive weight elements that share the same quantization scale $s$ and bias $b$, which control the quantization step size and zero point, respectively.

The weights $W$ within the quantization group are quantized according to the following equation:

$$\tilde{w} = R(clamp(\frac{W - b}{s}, Q_n, Q_p)), \quad W_q = \tilde{w} \times s + b \tag{3}$$

Here, $\tilde{w}$ denotes the quantized integer value obtained through a sequence of division, clamping, and rounding. Clamping is applied within the range $Q_n = -2^{n-1}$ to $Q_p = 2^{n-1} - 1$, where $n$ is the bit-width, followed by the rounding function $R$. $W_q$ represents the dequantized version of the quantized weight, which is adjusted using $s$ and $b$ from $\tilde{w}$ to approximate the original weight.

During QAT, the straight-through estimator (STE) approximates the derivative of the rounding function with an identity function (Bengio et al., 2013; Choi et al., 2018; Esser et al., 2020), enabling gradients to propagate through non-differentiable rounding operations and allowing effective weight parameter training. Building on conventional QAT, LSQ (Esser et al., 2020) and LSQ+ (Bhalgat et al., 2020) extend this process by training quantization parameters $s$ and $b$, alongside model weights. This tuning scheme provides finer control over quantization, improving model accuracy. The quantization parameters tuning during backpropagation proceeds as follows, where $w$ denotes the value $\frac{W-b}{s}$:

$$\frac{\partial L}{\partial s} = \frac{\partial W_q}{\partial s} \frac{\partial L}{\partial W_q} \quad s.t. \quad \frac{\partial W_q}{\partial s} = -w + \tilde{w}, \quad \text{if } Q_n \leq w \leq Q_p \tag{4}$$

$$\frac{\partial L}{\partial b} = \frac{\partial W_q}{\partial b} \frac{\partial L}{\partial W_q} \quad s.t. \quad \frac{\partial W_q}{\partial b} = 1, \quad \text{if } w < Q_n \text{ or } w > Q_p \tag{5}$$

More details on QAT with LSQ and LSQ+ are provided in Appendix A.1.

## 2.3 LLM QUANTIZATION

Quantization compress LLMs by lowering the bit precision of model parameters (Hubara et al., 2017). However, a key challenge with quantization is the introduction of errors that reduce model accuracy, leading to extensive research aimed at mitigating these losses through calibration or training. A notable examples of PTQ for LLM compression are GPTQ(Frantar et al., 2023) and OmniQuant (Shao et al., 2023). In contrast, QAT integrates quantization into the training process, adaptively training model parameters to account for quantization errors during training, ensuring that the quantized model retains much of its accuracy and functionality through training.

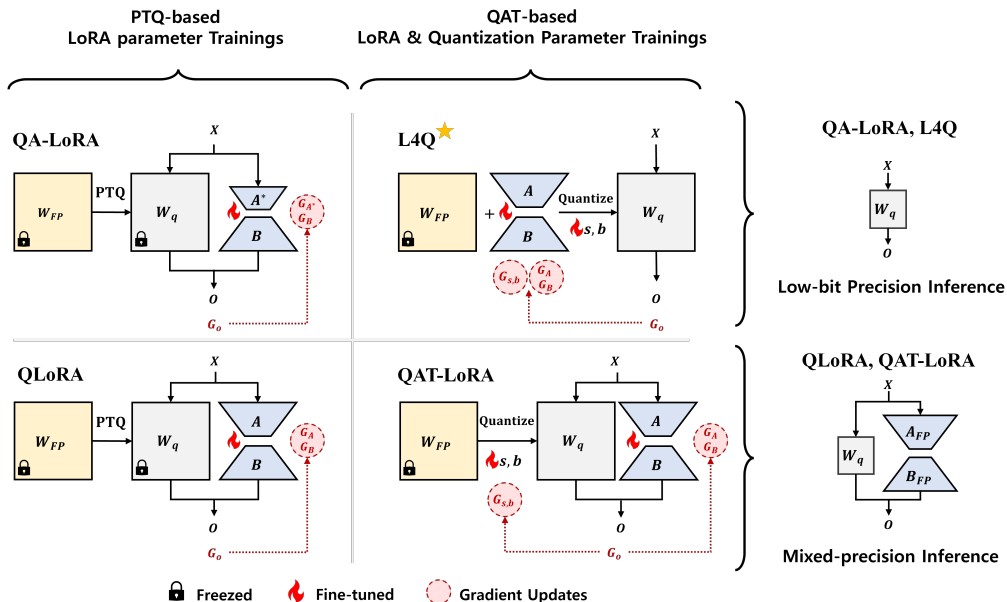

Figure 2: A categorization of training scheme and inference strategy of QA-LoRA, QLoRA, QAT-LoRA, and L4Q. Compared to QA-LoRA, L4Q utilizes higher optimization ability with non-constrained LoRA parameters and quantization parameters. Additionally, compared to QLoRA and QAT-LoRA, L4Q exploit fully-quantized weights rather than the mixed-precision weights during inference and perform a solid co-optimization of parameters.

Despite its advantages, QAT faces challenges, primarily due to its high training overhead, which limits its use in resource-intensive models like LLMs. The memory overhead of QAT stems from storing weight gradients and their optimizer states, each requiring multiple times the memory of the weights. For example, applying QAT to a 7B model that fine-tunes all weight parameters requires at least 14GB for the model weights, 14GB for the gradients, 28GB for optimizer states, and additional dozens of GB for activations. This often exceeds the memory capacity of a single GPU, which typically offers a maximum of 80GB of memory. Therefore, previous works have primarily focused on the application of PTQ for LLM compression, while research on QAT remains in its early stages, especially in terms of improving training efficiency.

## 2.4 QUANTIZATION-AWARE PEFT

To improve the accuracy of quantized LLMs, recent research has introduced quantization-aware PEFT approaches (Dettmers et al., 2024b; Kim et al., 2024; Xu et al., 2023; Li et al., 2024). Among these, QLoRA (Dettmers et al., 2024b), QA-LoRA (Xu et al., 2023), and LoftQ (Li et al., 2024) stand out as notable methods. As illustrated in Figure 2, QLoRA begins by applying PTQ to a pre-trained model. After this initial quantization, LoRA fine-tuning is performed, with the quantized weight parameters kept frozen. The LoRA parameters are fine-tuned using higher precision formats such as bfloat16 or float16, allowing the method to correct quantization errors during the fine-tuning. However, QLoRA introduces computational inefficiencies during inference due to the additional forward path on LoRA parameters. This inefficiency arises because the high-precision LoRA parameters and low-precision quantized weights cannot be merged into low-precision values. We further examine the impact of this unmerged LoRA path on inference efficiency by comparing the speed of fully-quantized models with mixed-precision models in Section 4.

QA-LoRA (Xu et al., 2023) addresses the issue of high-precision LoRA parameters by modifying the structure of the LoRA matrix, allowing these parameters to be integrated with the quantized weights after training (Figure 2). The input dimension of the LoRA matrix $A$ is set to the number of weight quantization groups. This adjustment ensures that each element of $BA$ corresponds directly with individual quantization groups, enabling the LoRA parameters to be seamlessly integrated into

the quantization bias as $b' = b - \alpha BA$ at the end of training. However, this solution presents a new challenge: the constrained LoRA structure in this setup limits the model's ability to achieve optimal accuracy during the PEFT stage.

A broader issue with previous quantization-aware PEFT approaches is that quantization and fine-tuning are performed sequentially rather than simultaneously. Starting the fine-tuning process from a pre-quantized model with inherent quantization errors is less optimal than starting from a pre-trained model. While LoftQ attempts to address these quantization errors by approximating them using LoRA with iterative singular-value decomposition (SVD), the limitations on subsequent adaptation persist, making it challenging to fully recover and optimize model performance during fine-tuning. These challenges highlight the need for continued research to improve quantization-aware PEFT techniques, aiming to enhance both the quantization and PEFT processes for better accuracy and efficiency in LLMs.

## 3    METHODS

### 3.1    STRAIGHTFORWARD INTEGRATION OF QAT AND LoRA

**QAT-LoRA**   One of the key principles in our proposed L4Q scheme is the integration of the QAT and LoRA to facilitate the simultaneous calibration for quantization and fine-tuning on downstream tasks. To achieve this, we begin with a straightforward integration of QAT and LoRA, referred to as QAT-LoRA, which serves as our baseline approach for combining QAT and PEFT. In QAT-LoRA, pre-trained weights are frozen, while LoRA parameters are added to the linear layers (Figure 2). Additionally, quantization scales and bias parameters are introduced, similar to advanced QAT techniques like LSQ, which are crucial for calibrating the quantization function. Freezing the weights reduces the need for optimizer states, while a small number of LoRA and quantization parameters are introduced to approximate updates to the weight matrix and to update the quantization function, respectively. This results in more efficient memory usage compared to standard QAT. Detailed analysis results of the memory efficiency of QAT-LoRA is further discussed in Section 4.

**Limitations of QAT-LoRA**   However, several issues arise with this straightforward integration of QAT and LoRA, where quantized weights and LoRA parameters remain distinct. First, although freezing the pre-trained weights eliminates the need for optimizer states, weight gradients $\frac{\partial L}{\partial W_q}$ must still be stored to update the quantization parameters, as shown in Equations 4- 5. As a result, QAT-LoRA still incurs memory overhead from weight gradients, undermining the memory efficiency benefits of LoRA fine-tuning. Secondly, QAT-LoRA produces a mixed-precision model with both quantized weights and high-precision LoRA parameters. This mixed-precision approach negates the advantages of LLM quantization, similar to previous methods such as QLoRA and LoftQ discussed in Section 2.4. Lastly, the gradient updates for quantization and LoRA parameters are decoupled, limiting the potential for comprehensive optimization across the model. As outlined in Equations 4- 5, updates to the quantization parameters rely on the quantized weight matrix $W_q$, while updates to the LoRA parameters depend on weights $A$ and $B$. This limits the effectiveness of model training, as it prevents holistic adjustments where changes in LoRA parameters could directly influence quantization adjustments and vice versa.

To address these challenges, we introduce L4Q, an enhanced integration of QAT and LoRA. L4Q features an advanced layer design that seamlessly integrates QAT and LoRA. By applying quantization after merging the weights and LoRA parameters, along with a custom backpropagation path that reduces the memory overhead from the complex quantization and LoRA processes, L4Q effectively overcomes the primary challenges encountered with QAT-LoRA.

### 3.2    L4Q: LOW-RANK ADAPTIVE LEARNING QUANTIZATION

**Fully-Quantized Linear Layer**   As high-precision LoRA weights introduces inference overhead, it is crucial to design a fully-quantized linear layer. In this context, L4Q first combines the original weights $W_0$ and the LoRA parameters $BA$ into a unified parameter matrix:

$$W_{comb} = W_0 + \alpha BA \qquad (6)$$

Then, quantization is applied to the fully combined weight $W_{comb}$ as follows:

$$W_q = R(clamp(\frac{W_{comb} - b}{s}, Q_n, Q_p)) \times s + b \tag{7}$$

In this way, during inference, L4Q only uses quantized weights, simplifying the forward path of the linear layer as follows:

$$Y = W_q X \tag{8}$$

While QA-LoRA also achieves fully-quantized linear layers by introducing constraints on the LoRA parameter structure, the proposed L4Q imposes no such restrictions. This flexibility allows L4Q to fully leverage the benefits of LoRA-based fine-tuning, all while achieving fully-quantized linear layers.

**Memory Efficient QAT**  As discussed in the previous section, QAT requires weight gradients for training the quantization parameters $s$ and $b$. Since weight gradients are a major source of memory overhead during training, we compute these gradients locally in the backpropagation path of the linear layer, utilizing the input and output of the layer. The weight gradients are calculated as follows:

$$\frac{\partial L}{\partial W_q} = \frac{\partial L}{\partial Y} X^\top \tag{9}$$

We use these weight gradients to calculate gradients of $s$ and $b$ with Equation 4 and 5. Once the gradient computation for the linear layer is complete, the weight gradients are immediately flushed to conserve memory.

**Efficient LoRA Training**  To compute the gradients of the LoRA parameters in the L4Q linear layer, we must trace back from Equation 8 to 6. Due to the full integration of quantization and LoRA parameters (Equation 6 and 7), the gradients of the LoRA parameters also rely on the weight gradients. This then be described as follows:

$$\frac{\partial L}{\partial A} = \frac{\partial L}{\partial W_q} \frac{\partial W_q}{\partial A}, \quad \frac{\partial L}{\partial B} = \frac{\partial L}{\partial W_q} \frac{\partial W_q}{\partial B} \tag{10}$$

Since we reuse the weight gradient $\frac{\partial L}{\partial W_q}$ that have already been computed for QAT, we only need to compute $\frac{\partial W_q}{\partial A}$ and $\frac{\partial W_q}{\partial B}$ to obtain the gradients of LoRA parameters. Both terms are derived by applying the chain rule to Equation 7 and Equation 6. Since Equation 7 contains a rounding function, we apply STE and clamping with conditional gradient propagation to $\frac{\partial W_q}{\partial A}$ and $\frac{\partial W_q}{\partial B}$. Moreover, in Equation 6, the LoRA parameters $A$ and $B$ are simply multiplied, so its gradients are expressed directly as $\alpha B^\top$ and $\alpha A^\top$, respectively. This leads to the following expressions for $\frac{\partial W_q}{\partial A}$ and $\frac{\partial W_q}{\partial B}$:

$$\frac{\partial W_q}{\partial A} = \begin{cases} \alpha B^\top, & \text{if } Q_n \leq w \leq Q_p \\ 0, & otherwise \end{cases}, \quad \frac{\partial W_q}{\partial B} = \begin{cases} \alpha A^\top, & \text{if } Q_n \leq w \leq Q_p \\ 0, & otherwise \end{cases} \tag{11}$$

Therefore, the proposed L4Q efficiently processes LoRA training by simply reusing the weight gradients computed for QAT parameter training. For more detailed explanations of the gradient calculation in L4Q, please refer to Appendix A.2, and the memory efficiency of L4Q will be further examined in Section 4.

**Joint Optimization of Quantization and LoRA parameters**  Since $\frac{\partial L}{\partial W_q}$ is involved in the gradient calculation for the LoRA parameters (Equation 10), the proposed L4Q ensures that the impact of quantization is directly reflected in the updates to the LoRA parameters. This enables the joint optimization of LoRA parameters and the quantization process, enhancing the accuracy of the fully-quantized LLMs.

In summary, the proposed L4Q produces a fully-quantized model for memory-efficient and fast LLM inference by fully integrating the model weights and LoRA parameters prior to the quantization process. Additionally, the training process of L4Q is memory-efficient due to careful handling of gradient computation for quantization. Finally, L4Q can improve the accuracy of quantized LLMs through the joint optimization of the quantization and LoRA parameters.

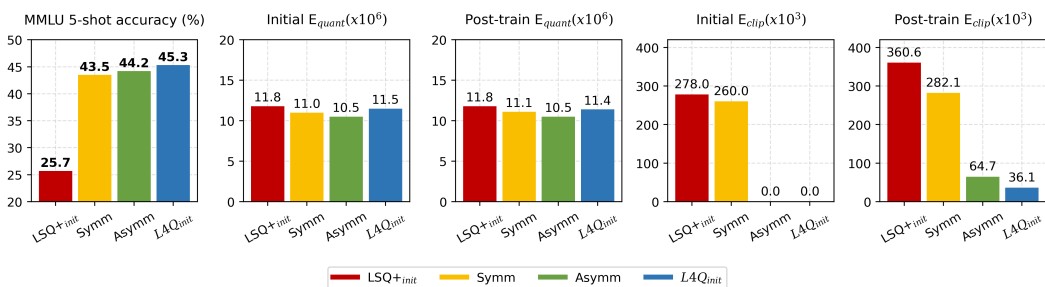

Figure 3: The MMLU 5-shot benchmark results for LLaMA-2 7B models with 100-step training, along with the sum of quantization and clipping errors, are presented for the initialization methods: LSQ+, symmetric, asymmetric, and L4Q. Errors are reported at both the initial point (Initial) and after training (Post-train).

### 3.3 QUANTIZATION PARAMETER INITIALIZATION

In L4Q, we address the outlier-sensitive nature of LLMs by improving the quantization parameter initialization at the start of fine-tuning. While the initialization scheme in LSQ+ (LSQ+$_{init}$) has been shown to be effective by using multiples of standard deviation of weights as the quantization range and setting the quantization scale accordingly, this approach was originally designed for CNNs, not transformer-based LLMs. Studies on LLM quantization have shown that outliers in activation and their corresponding salient weights significantly impact model performance (Xiao et al., 2023; Dettmers et al., 2024a; Lin et al., 2024). In LLM, the standard deviation of the small number of weight groups is often much smaller than the absolute maximum value, causing large weights to be more prone to clipping, resulting in significant clipping errors. To address this, we introduce a quantization initialization scheme, L4Q$_{init}$, which minimizes clipping errors by adopting a conservative quantization scale that captures both minimum and maximum outliers during training. The detailed equation is as follows:

$$s = Max(|\frac{Min(W)}{Qn}|, |\frac{Max(W)}{Qp}|), \quad b = 0 \tag{12}$$

Note that conventional symmetric initialization involves division of the absolute maximum value of the numbers, and asymmetric initialization involves division of the minimum to maximum range of the number for the quantization scale initialization. Detailed explanation on LSQ+, symmetric, asymmetric, and L4Q quantization initialization is presented in Appendix B.

We evaluate the models trained with L4Q using various initialization methods, comparing the accuracy, the sum of quantization error $|W - W_q|$, and the sum of clipping error of the overflowed outliers at both the initialization point and the end of training. As shown in Figure 3, L4Q$_{init}$ achieves the highest accuracy, while LSQ+$_{init}$ struggles to recover from quantization errors. Although the overall quantization error remains similar across methods, clipping errors are notably higher in LSQ+$_{init}$ and symmetric initialization. Additionally, while asymmetric initialization avoids outliers at the start, it fails to handle emerging outliers during fine-tuning as the weight distribution shifts. Its tight initialization, focused on the minimum to maximum range, becomes insufficient as weights change, leading to higher clipping error. In contrast, L4Q$_{init}$ considers the broader weight distribution in LLMs, reducing both quantization and clipping errors effectively.

### 3.4 OVERALL L4Q ALGORITHM

The overall L4Q algorithm is outlined in Algorithm 1. In the initial phase, L4Q undergoes a warm-up stage with LoRA fine-tuning only. Since the quantization function includes a clamping mechanism, applying it to LoRA at the beginning of training can suppress gradient updates for outlier values, which negatively affects LLM accuracy. Hence, L4Q warms up the LoRA parameters before applying quantization. The LoRA warm-up consists of $T_{wp}$ iterations, where $T_{wp} << T$, with $T$ representing the total number of training steps. An analysis of the effect of LoRA warm-up can be found in Appendix C. After the warm-up phase, L4Q combines the pre-trained weights and

---

**Algorithm 1** L4Q Algorithm

---

**Require:** Pre-trained weight $W_0$, Fine-tuned weight $W_{comb}$, Quantized weight $W_q$
**Require:** LoRA parameter $A$, $B$, LoRA configuration $r, \alpha$
**Require:** Quantization parameter $s, b$, Quantization bit-width $n$
**Require:** Training step $T$, LoRA warm-up step $T_{wp}$, State $I_{init}$
   Initialize $A \sim N(0, \sigma^2), B \leftarrow 0, \alpha \leftarrow \frac{1}{2r}, I_{init} \leftarrow False$
   **for** $t = 1$ to $T$ **do**
      **if** $t \leq T_{wp}$ **then**                                 ▷ LoRA warm-up stage
         Forward $W_0 X + \alpha BAX$ and Update $(A, B)$
      **else**                                           ▷ L4Q fine-tuning stage
         **if** not $I_{init}$ **then**
            Set $W_{comb} \leftarrow W_0 + \alpha BA$
            Initialize $s, b \leftarrow$ L4Q$_{init}$           ▷ L4Q$_{init}$ described in the equation 12
            Set $I_{init} \leftarrow$ True
         **end if**
         Forward $W_q X$ and Update $(A, B, s, b)$         ▷ $W_q$ described in the equation 8
      **end if**
   **end for**

---

LoRA parameters (Equation 6), and then initializes the quantization parameters $s$ and $b$, which determine the quantization range, using these combined weights. While previous QAT schemes use the standard deviation of weights to set the initial quantization range, and prior LLM quantization approaches rely on the min/max of the weights, L4Q employs a relaxed min/max-based initialization as discussed in Section 3.3. Then, the proposed L4Q conducts both QAT and LoRA, with gradient computation as described in Section 3.2, to achieve high-accuracy fully-quantized models.

## 4 EXPERIMENTS

### 4.1 EXPERIMENTAL SETTINGS

**Target Foundation Models** OpenLLaMA[1], LLaMA-1 (Touvron et al., 2023a), and LLaMA-2 (Touvron et al., 2023b) model families are used for the evaluation. Specifically, we assess the OpenLLaMA model with 3B parameters, LLaMA-1 models with 7B, 13B, and 33B parameters, and LLaMA-2 models with 7B and 13B parameters. When applying quantization, the quantization group size is set to 128 for the LLaMA-1 and LLaMA-2 models, and 64 for the OpenLLaMA models.

**Baselines** We compare the proposed L4Q with previous PTQ and QA-PEFT methods. The baseline PTQ methods include GPTQ and OmniQuant, while the baseline QA-PEFT methods include QLoRA, QA-LoRA, and LoftQ. Although the original QLoRA and LoftQ employ non-uniform quantization, we apply uniform quantization in our experiments for consistency across methods. These uniformly quantized versions are referred to as QLoRA* and LoftQ*.

**Evaluation Setups** We establish the L4Q framework based on the Lit-GPT[2] and huggingface transformers[3], both of which are open-source LLM frameworks. The rank of LoRA parameter $r$ is set to 4 by default, as our experiments showed that $r = 4$ is sufficient to maintain performance. Further details on the effect of rank size can be found in Appendix D. Meanwhile, as QA-LoRA reduces the input dimension of the LoRA parameters to integrate them with the quantization bias parameter, we double the rank $r$ to 8 for a fair comparison in terms of the number of LoRA parameters. For the fine-tuning process, we use the Stanford-Alpaca (Taori et al., 2023) dataset that consists of 52k instruction-following samples generated from the instruction-tuned GPT 3.5 model (Brown et al., 2020). This dataset is split into a training set with 50k samples and a validation set with 2k samples. In line with standard LLM training conventions, we used bfloat16 for numerically stable fine-tuning. The batch size for fine-tuning is 128. All experiments were conducted on an NVIDIA A100 80GB GPU. Detailed hyperparameter settings for fine-tuning are provided in Appendix E.

---

[1] https://github.com/openlm-research/open_llama
[2] https://github.com/Lightning-AI/lit-gpt.git
[3] https://github.com/huggingface/transformers.git

| Methods | OpenLLaMA | | LLaMA-1 | |
|---|---|---|---|---|
| | 3B | 7B | 13B | 33B |
| LoRA | 15.1 | 25.1 | 43.8 | 71.9 |
| QLoRA | 5.2 | 7.9 | 19.6 | 31.9 |
| LoftQ | 5.2 | 7.9 | 19.6 | 31.9 |
| QA-LoRA | 7.8 | 14.8 | 27.8 | 67.2 |
| QAT | 44.2 | 79.5 | OOM | OOM |
| QAT-LoRA | 22.6 | 41.9 | 70.6 | OOM |
| L4Q | 15.3 | 25.4 | 44.3 | 73.2 |

Table 1: Memory cost (GB) for fine-tuning LLMs on NVIDIA A100 GPU. (OOM: Out of Memory)

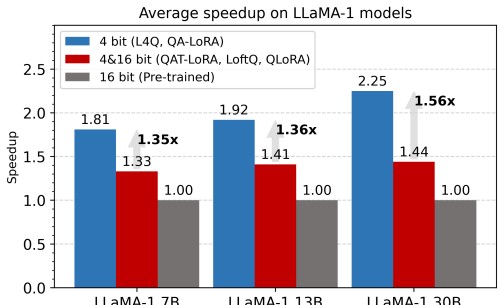

Figure 4: The average inference speedup of quantized models compared to pre-trained models.

**Evaluation Metrics** We evaluate the accuracy of LLMs on Commonsense QA (CSQA) (Gao et al., 2021) and MMLU (Hendrycks et al., 2021) benchmarks. The CSQA benchmark includes tasks from Hellaswag (Zellers et al., 2019), PIQA (Bisk et al., 2020), ARC-challenge and ARC-easy (Clark et al., 2018), Winogrande (Sakaguchi et al., 2020), BoolQ (Clark et al., 2019), and Openbook QA (Talmor et al., 2019). The MMLU benchmark spans four subject categories: Humanities, STEM, Social Sciences, and others made up of 57 subcategories of language tasks.

## 4.2 Evaluation Results

**Memory Cost for Fine-Tuning** We measure the peak memory usage during fine-tuning of 4-bit LLMs using both the baseline methods and L4Q, as reported in Table 1. Since PTQ-based PEFT methods apply PTQ before fine-tuning, the size of linear layers in the model is reduced by a factor of 1/4, resulting in lower memory usage compared to naïve LoRA, which fine-tunes full-precision models. While both QAT and QAT-LoRA result in 2–3 times higher memory costs compared to LoRA, the memory usage of L4Q is similar to that of LoRA. Although L4Q cannot reduce memory costs as much as PTQ-based PEFT methods due to the use of 16-bit full-precision models during fine-tuning, this result highlights that L4Q is well-designed to leverage the memory-efficient nature of LoRA in training. Further analysis on the training efficiency of L4Q and baseline methods can be found in Appendix F.

**Inference Speedup** We measure the inference speed of 16-bit pre-trained models and quantized models using LLaMA-1 models. The quantized models include fully-quantized 4-bit models (L4Q and QA-LoRA), which contain only quantized parameters, and mixed-precision 4&16-bit models (LoftQ*, QLoRA*, and QAT-LoRA), which use additional 16-bit LoRA parameters. The inference speed was measured with input batch sizes ranging from 1 to 64. The average speedup of quantized models compared to full-precision 16-bit models is reported in Figure 4. The 4-bit models achieve a speedup of 1.8× to 2.3× over the 16-bit models. More importantly, these 4-bit models achieve a 1.4× to 1.6× speedup compared to mixed-precision 4&16-bit models, which are also quantized versions of LLMs. This demonstrates that the full integration of QAT and LoRA in L4Q plays a crucial role in achieving the best inference speed. More analysis on speedup can be found in Appendix G.

**Accuracy Results** We compare the accuracy of baselines and L4Q. Specifically, we evaluate zero-shot accuracy on the CSQA benchmark, and evaluate both zero-shot and few-shot (5-shot) accuracy on the MMLU benchmark. Table 2 presents a comprehensive comparison between baselines and the proposed L4Q after 4-bit LLM quantization. Since previous QA-PEFT methods involve a fine-tuning stage after PTQ, they generally achieve higher accuracy compared to PTQ methods. L4Q further improves accuracy of 4-bit models by adopting QAT strategy, and it achieves accuracy comparable to 16-bit models. Moreover, L4Q consistently outperforms QAT-LoRA, which adopts QAT but keeps quantization and LoRA parameters decoupled. This highlights the advantage of L4Q in achieving superior accuracy through the joint optimization of quantization and LoRA parameters. We also evaluate accuracy with 3-bit LLM quantization. As presented in Table 3. L4Q achieves the best accuracy in this case as well. Notably, the impact of applying QAT on accuracy is more pronounced in 3-bit quantization, as previous approaches experience significant accuracy degradation with 3-bit quantization.

Table 2: Accuracy (%) evaluation results with 4-bit quantization. The bit precision of weight parameters is indicated under the method name. The notation '4+16' refers to the requirement of 16-bit LoRA parameters alongside 4-bit weights for inference.

| Model | Benchmark | Pre-trained 16 | LoRA 16 | GPTQ 4 | OmniQ. 4 | LoftQ* 4+16 | QLoRA* 4+16 | QA-LoRA 4 | QAT-LoRA 4+16 | L4Q 4 |
|---|---|---|---|---|---|---|---|---|---|---|
| OpenLLaMA 3B | CSQA | 54.8 | 55.9 | 50.7 | 54.1 | 54.2 | 54.4 | 54.5 | 54.6 | **55.0** |
| LLaMA-1 7B | CSQA | 61.7 | 63.4 | 59.4 | 58.1 | 62.6 | 61.3 | 61.3 | 62.4 | **62.7** |
| | $MMLU_{0-shot}$ | 32.5 | 36.3 | 28.3 | 30.9 | 33.0 | 32.8 | 34.5 | 33.8 | **34.9** |
| | $MMLU_{5-shot}$ | 35.1 | 36.7 | 32.7 | 33.3 | 35.1 | 33.6 | 35.6 | 34.8 | **35.7** |
| LLaMA-1 13B | CSQA | 63.8 | 65.2 | 63.5 | 60.4 | 64.2 | 63.8 | 62.5 | 64.4 | **64.5** |
| | $MMLU_{0-shot}$ | 43.6 | 44.3 | 40.1 | 42.6 | 42.4 | 42.1 | 42.4 | 42.0 | **43.2** |
| | $MMLU_{5-shot}$ | 46.3 | 47.0 | 45.7 | 45.7 | 45.4 | 45.9 | 45.8 | 45.5 | **46.0** |
| LLaMA-1 33B | CSQA | 67.4 | 68.3 | 65.7 | 62.9 | 67.4 | 66.2 | 65.3 | 67.3 | **67.5** |
| | $MMLU_{0-shot}$ | 53.0 | 54.4 | 51.4 | 52.0 | 51.8 | 51.0 | 48.9 | 52.3 | **53.3** |
| | $MMLU_{5-shot}$ | 56.4 | 57.6 | 55.7 | 55.8 | 56.4 | 55.6 | 55.0 | 56.7 | **56.7** |
| LLaMA-2 7B | CSQA | 61.9 | 63.3 | 60.7 | 59.5 | 61.7 | 61.3 | 61.0 | 61.9 | **63.6** |
| | $MMLU_{0-shot}$ | 41.6 | 43.9 | 37.1 | **41.0** | 38.5 | 38.6 | 38.9 | 37.9 | 40.9 |
| | $MMLU_{5-shot}$ | 45.4 | 46.0 | 42.9 | 45.4 | 43.7 | 44.6 | 44.4 | 43.8 | **45.5** |
| LLaMA-2 13B | CSQA | 65.0 | 66.5 | 64.4 | 59.9 | 64.9 | 64.0 | 64.5 | 64.7 | **65.8** |
| | $MMLU_{0-shot}$ | 52.1 | 52.5 | 50.0 | 51.8 | 51.7 | 50.7 | 50.4 | 50.7 | **51.9** |
| | $MMLU_{5-shot}$ | 54.8 | 55.7 | 54.7 | 54.7 | 54.5 | 54.2 | 54.1 | 53.8 | **55.2** |

Table 3: Accuracy (%) evaluation results with 3-bit quantization. The bit precision of weight parameters is indicated under the method name. The notation '3+16' refers to the requirement of 16-bit LoRA parameters alongside 3-bit weights for inference.

| Model | Benchmark | Pre-trained 16 | LoRA 16 | GPTQ 3 | OmniQ 3 | LoftQ* 3+16 | QLoRA* 3+16 | QA-LoRA 3 | QAT-LoRA 3+16 | L4Q 3 |
|---|---|---|---|---|---|---|---|---|---|---|
| OpenLLaMA 3B | CSQA | 54.8 | 55.9 | 52.2 | 50.0 | 38.1 | 51.0 | 51.5 | 53.2 | **54.0** |
| LLaMA-1 7B | CSQA | 61.7 | 63.4 | 53.4 | 56.5 | 49.8 | 59.1 | 58.7 | 60.7 | **61.2** |
| | $MMLU_{0-shot}$ | 32.5 | 36.3 | 23.7 | 29.0 | 23.4 | 27.7 | 28.0 | 30.6 | **30.6** |
| | $MMLU_{5-shot}$ | 35.1 | 36.7 | 27.3 | 31.6 | 23.1 | 31.5 | 29.1 | 31.5 | **31.8** |
| LLaMA-1 13B | CSQA | 63.8 | 65.2 | 61.0 | 58.9 | 54.0 | 61.3 | 61.1 | 63.2 | **63.4** |
| | $MMLU_{0-shot}$ | 43.6 | 44.3 | 33.1 | 34.8 | 25.0 | 36.1 | 37.5 | 38.8 | **40.7** |
| | $MMLU_{5-shot}$ | 46.3 | 47.0 | 38.2 | 41.6 | 25.3 | 40.4 | 38.2 | 40.9 | **41.8** |
| LLaMA-2 7B | CSQA | 61.9 | 63.3 | 57.6 | 57.9 | 34.7 | 57.6 | 56.3 | 57.4 | **61.3** |
| | $MMLU_{0-shot}$ | 41.6 | 43.9 | 31.3 | 34.3 | 22.9 | 32.5 | 31.0 | 31.5 | **34.9** |
| | $MMLU_{5-shot}$ | 45.4 | 46.0 | 37.5 | 37.7 | 24.2 | 37.6 | 37.5 | 36.8 | **38.0** |
| LLaMA-2 13B | CSQA | 65.0 | 66.5 | 61.7 | 59.9 | 39.3 | 62.5 | 61.7 | 64.3 | **65.1** |
| | $MMLU_{0-shot}$ | 52.1 | 52.5 | 46.3 | 46.3 | 23.5 | 46.8 | 46.4 | 45.9 | **47.1** |
| | $MMLU_{5-shot}$ | 54.8 | 55.7 | 50.4 | 50.2 | 26.0 | **50.6** | 49.9 | 48.9 | 50.0 |

## 5 CONCLUSION

In this work, we introduce L4Q, a parameter-efficient quantization-aware fine-tuning method for large language models. L4Q enables element-wise adaptation of model weights for downstream tasks while simultaneously optimizing quantization parameters. This concurrent optimization ensures that the adaptation parameters effectively account for quantization errors. We demonstrate the efficiency of L4Q, which significantly reduces training resource requirements compared to traditional QAT. Moreover, since the L4Q layer is designed to produce fully quantized low-bit model weights, it maintains inference efficiency, unlike QLoRA, LoftQ, or QAT-LoRA, which result in mixed-precision models. The effectiveness of L4Q as a QAT framework is further supported by experimental results across various task evaluations. L4Q consistently achieves superior quality maintenance on language tasks, demonstrating its enhanced adaptability compared to QAT-LoRA and PTQ-based PEFT methods.

## ETHICS STATEMENTS

This paper presents work aimed at advancing the field of machine learning, particularly in the context of language models. While our research contributes to the development and exploitation of LLMs, we acknowledge that there are potential societal consequences associated with this work. However, we do not delve into the potential harms or biases inherent in these models.

## REPRODUCIBILITY STATEMENT

This paper introduces a novel algorithm for quantization layer design and the training process. To ensure reproducibility, the source code is provided as an anonymous, downloadable link in the supplementary materials, along with detailed explanations of the environmental setup and execution process. The theoretical results are thoroughly explained with clear assumptions, and complete proofs are included in the appendix. Additionally, detailed descriptions of the data processing steps for all datasets used in the experiments are available in the supplementary materials. These resources are intended to fully enable the reproduction of our results.

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

# A   DETAILS ON QUANTIZATION SCALE LEARNING PROCEDURE

## A.1   QUANTIZATION SCALE UPDATE ON QAT

From the conditions and notations in Equation 3, Equations 4 and 5 are derived as follows. First, the derivative of $s$ is presented as follows.

$$\frac{\partial W_q}{\partial s} = \frac{\partial}{\partial s}(\tilde{w} \times s + b) = s\frac{\partial}{\partial s}(\tilde{w}) + \tilde{w} = s\frac{\partial}{\partial s}(R \cdot clamp(w)) + \tilde{w} \tag{13}$$

By applying the STE, the rounding function $R$ is considered as an identity function. Therefore, the rounding function, combined with a clamp function $\tilde{R} := R \cdot clamp$, and its derivative is induced as follows. Note that $w = \frac{W-b}{s}$.

$$\tilde{R}(w) = \begin{cases} Qn, & if \ w < Qn \\ w, & if \ Q_n \leq w \leq Q_p \\ Qp, & if \ w > Q_p \end{cases} \quad \frac{\partial}{\partial w}\tilde{R}(w) = \begin{cases} 1, & if \ Q_n \leq w \leq Q_p \\ 0, & otherwise \end{cases} \tag{14}$$

By applying the chain rule, the derivation of term $R \cdot clamp(w) = \tilde{R}((W - b)/s)$ is expressed as below.

$$\frac{\partial}{\partial s}(\tilde{R}(w)) = \frac{\partial \tilde{R}}{\partial w}\frac{\partial w}{\partial s} = \frac{\partial \tilde{R}}{\partial w}\frac{\partial}{\partial s}(\frac{W-b}{s}) = \frac{\partial \tilde{R}}{\partial w}(-\frac{W-b}{s^2}) \tag{15}$$

Therefore, Equation 13 can be represented with a value $w$ and quantized value $\tilde{w}$ as follows.

$$\frac{\partial W_q}{\partial s} = s\frac{\partial \tilde{R}}{\partial w}(-\frac{W-b}{s^2}) + \tilde{w} = \frac{\partial \tilde{R}}{\partial w}(-\frac{W-b}{s}) + \tilde{w} = \begin{cases} Qn, & if \ w < Qn \\ -w + \tilde{w}, & if \ Q_n \leq w \leq Q_p \\ Qp, & if \ w > Q_p \end{cases} \tag{16}$$

Secondly, with a similar context above, the derivative of $b$ is presented as follows.

$$\frac{\partial W_q}{\partial b} = \frac{\partial}{\partial b}(\tilde{w} \times s + b) = s\frac{\partial}{\partial b}(\tilde{R}(w)) + 1 = s\frac{\partial \tilde{R}}{\partial w}(\frac{\partial}{\partial b}(\frac{W-b}{s})) + 1 \tag{17}$$

$$= \frac{\partial \tilde{R}}{\partial w}(-1) + 1 = \begin{cases} 0, & if \ Q_n \leq w \leq Q_p \\ 1, & otherwise \end{cases} \tag{18}$$

We also note that the gradient of $W_q$ is presented as follows.

$$\frac{\partial L}{\partial W_q} = \frac{\partial L}{\partial Y}X^\top \tag{19}$$

As a result, the updates on the quantization scale and bias are calculated as multiplication of Equation 19 with Equation 13 and with Equation 17, respectively. This update helps calibrate the quantization function, effectively reducing quantization errors.

## A.2 Quantization scale and LoRA Parameter update on L4Q

In L4Q, as described in Equation 8, the quantized weight $W_q$ is obtained as follows. First, the pre-trained model weight $W_0$ and LoRA parameters are integrated to $W_{comb} = W_0 + \alpha BA$. Next, the integrated weight is quantized by the quantization parameters $s, b$.

Here, the LoRA parameters A and B are independent of the quantization parameters, scale $s$ and bias $b$. Therefore, the derivatives of $s, b$ follow the same process as in Equation A.1, but with the term $w, \tilde{w}$ defined as follows. Note that $W_q = \tilde{w} \times s + b$.

$$w = \frac{W_0 + \alpha BA - b}{s}, \quad \tilde{w} = \tilde{R}(w) \quad s.t. \quad \tilde{R} = R \cdot clamp \tag{20}$$

Seen from the L4Q layer that integrates the LoRA parameters and quantization parameters together, $A, B$ are now considered as variables of $W_q$. Therefore, from the conditions in Equation 8, the derivative of $A, B$ is presented as follows.

$$\frac{\partial L}{\partial A} = \frac{\partial W_q}{\partial A}\frac{\partial L}{\partial W_q}, \quad \frac{\partial L}{\partial B} = \frac{\partial L}{\partial W_q}\frac{\partial W_q}{\partial B} \tag{21}$$

The derivatives $\frac{\partial w}{\partial A}$ and $\frac{\partial w}{\partial B}$ are then can be computed by applying the chain rule with $w$, as follows:

$$\frac{\partial W_q}{\partial A} = \frac{\partial w}{\partial A}\frac{\partial W_q}{\partial w}, \quad \frac{\partial W_q}{\partial B} = \frac{\partial W_q}{\partial w}\frac{\partial w}{\partial B} \tag{22}$$

From Equation 20, the terms $\frac{\partial w}{\partial A}, \frac{\partial w}{\partial A}$, and $\frac{\partial W}{\partial w}$ can be expressed as follows:

$$\frac{\partial w}{\partial A} = \frac{\alpha B^\top}{s}, \quad \frac{\partial w}{\partial B} = \frac{\alpha A^\top}{s}, \quad \frac{\partial W}{\partial w} = \frac{\partial}{\partial w}(\tilde{R}(w)s + b) = s\frac{\partial \tilde{R}}{\partial w} \tag{23}$$

Therefore, by substitution of Equation 23 and applying STE on $\frac{\partial \tilde{R}}{\partial w}$ from Equation 14 on Equation 22, the equation is simplified by the crossed-out products between the terms. As a result, the partial derivatives presented in Equation 10 can be derived as follows.

$$\frac{\partial W_q}{\partial A} = \frac{\partial w}{\partial A}\frac{\partial W_q}{\partial w} = (\frac{\alpha B^\top}{s})(\frac{s\partial \tilde{R}}{\partial w}) = \begin{cases} \alpha B^\top, & if \ Q_n \leq w \leq Q_p \\ 0, & otherwise \end{cases} \tag{24}$$

$$\frac{\partial W_q}{\partial B} = \frac{\partial W}{\partial w}\frac{\partial w}{\partial B} = (s\frac{\partial \tilde{R}}{\partial w})(\frac{\alpha A^\top}{s}) = \begin{cases} \alpha A^\top, & if \ Q_n \leq w \leq Q_p \\ 0, & otherwise \end{cases} \tag{25}$$

Finally, substitution of Equation 24 and 25 to Equation 21 derives the Equation 26 and 27.

$$\frac{\partial L}{\partial A} = \begin{cases} \alpha B^\top(\frac{\partial L}{\partial Y}X^\top), & if \ Q_n \leq w \leq Q_p \\ 0, & otherwise \end{cases} \tag{26}$$

$$\frac{\partial L}{\partial B} = \begin{cases} \alpha(\frac{\partial L}{\partial Y}X^\top)A^\top, & if \ Q_n \leq w \leq Q_p \\ 0, & otherwise \end{cases} \tag{27}$$

This form closely resembles the original backpropagation structure of the LoRA parameters $A, B$ as shown in Equation 2, where the updates are expressed as $\frac{\partial L}{\partial A} = \alpha\frac{\partial L}{\partial \tilde{X}}X^\top = \alpha(B^\top\frac{\partial L}{\partial Y})X^\top$, and $\frac{\partial L}{\partial B} = \alpha\frac{\partial L}{\partial Y}\tilde{X}^\top = \alpha\frac{\partial L}{\partial Y}(AX)^\top$, respectively. However, in L4Q, this process includes an added gating condition on the quantized weights, which accounts for the integration of quantization into the LoRA parameters. As a result, we conclude that the backward process of the L4Q layer, which integrates both quantization parameter learning and LoRA parameter adaptation, is designed to account for the impact of quantization on the LoRA parameter updates.

# B  QUANTIZATION INITIALIZATION

We evaluate various quantization initialization schemes within L4Q, including method introduced in Section 3.3, LSQ+ (Bhalgat et al., 2020), and conventional symmetric and asymmetric quantization parameter initialization. The methods are depicted as $L4Q_{init}$, $LSQ+_{init}$, Symm, Asymm, respectively. In specific, each methods can be represented as the equations below, with quantization scale $s$ and bias $b$ and group-wise aligned model weight $W$ with quantization bit-width $n$ and $Q_n = -2^{n-1}, Q_p = 2^{n-1} - 1$.

$$LSQ+_{init}: \quad s = \frac{Max(|\mu - 3\sigma(W)|, |\mu + 3\sigma(W)|)}{2^{n-1}} \tag{28}$$

$$b = 0 \tag{29}$$

$$Symm: \quad s = \frac{Max(Abs(W))}{2^{n-1}} \tag{30}$$

$$b = 0 \tag{31}$$

$$Asymm: \quad s = \frac{Max(W) - Min(W)}{Q_p - Q_n} \tag{32}$$

$$b = Max(W) - s \times Q_p = Min(W) - s \times Q_n \tag{33}$$

$$L4Q_{init}: \quad s = Max(|\frac{Min(W)}{Qn}|, |\frac{Max(W)}{Qp}|) \tag{34}$$

$$b = 0 \tag{35}$$

We report the detailed model accuracy evaluation results of L4Q fine-tuning across different initialization methods, along with the quantization error and clipping error for each method, measured both at the initialization point and the end of the training, in Table 4. The LLaMA-2 7B model was trained for a total of 12,800 iterations with batch size 128, using the same hyperparameters as those in the main evaluation.

Table 4: MMLU 5-shot benchmark and the sum of quantization errors for various quantization parameter initialization methods within L4Q on the LLaMA-2 7B model. Quantization errors are represented in order of $10^6$ and clipping errors are represented in order of $10^3$.

| Model | Method | #Bits | MMLU 5-shot | | | | | Initial | | Post-train | |
|---|---|---|---|---|---|---|---|---|---|---|---|
| | | | Human. | STEM | Social. | Others | Average | $E_{quant}$ | $E_{clip}$ | $E_{quant}$ | $E_{clip}$ |
| LLaMA-2 7B | LSQ+ | 4 | 26.7 | 26.8 | 26.2 | 22.9 | 25.7 | 11.8 | 278.0 | 11.8 | 360.6 |
| | Symm | 4 | 40.8 | 35.9 | 48.2 | 50.1 | 43.5 | 11.1 | 260.0 | 11.0 | 282.1 |
| | Asymm | 4 | 41.0 | 37.1 | 49.7 | 50.2 | 44.2 | 10.5 | 0.0 | 10.5 | 64.7 |
| | L4Q | 4 | 42.9 | 37.7 | 50.5 | 51.9 | 45.3 | 11.4 | 0.0 | 11.6 | 36.1 |

# C    EFFECT OF LoRA WARM-UP ON L4Q TRAINING

We investigated the effect of LoRA warm-up on L4Q training and found that a small number of LoRA warm-up steps is often beneficial for quickly restoring model accuracy. Using LLaMA-1 and LLaMA-2 7B models, we trained the models for a total of 12,800 iterations with 128 batches. The remaining training conditions, such as the quantization target layers and the rank of LoRA parameters, were kept consistent with the main experimental setup. The Commonsense QA evaluation results are presented in Table 5.

Table 5: Commonsense QA benchmark result with the variations of LoRA warm-up on LLaMA-1 and LLaMA-2 7B models. The numbers represent the accuracy(%) of each task.

| Model | #Bits | Warmup | Hella. | PIQA | ARC-c | ARC-e | Winog. | BoolQ | OBQA | Avg. |
|---|---|---|---|---|---|---|---|---|---|---|
| LLaMA-1 7B | 4 | **0** | 58.0 | 78.9 | 45.1 | 76.6 | 70.1 | 76.5 | 37.0 | **63.2** |
| | | 10 | 58.0 | 78.5 | 44.9 | 76.4 | 69.7 | 76.5 | 36.4 | 62.9 |
| | | 20 | 57.9 | 78.8 | 45.2 | 76.4 | 70.1 | 76.1 | 36.2 | 62.9 |
| | | 40 | 57.5 | 78.6 | 45.6 | 76.0 | 69.8 | 75.8 | 35.4 | 62.7 |
| | | 80 | 57.2 | 78.6 | 45.1 | 75.3 | 69.6 | 75.1 | 34.8 | 62.2 |
| | 3 | 0 | 55.8 | 77.2 | 41.9 | 73.6 | 68.1 | 76.1 | 32.8 | 60.8 |
| | | **10** | 55.7 | 77.2 | 42.0 | 74.0 | 68.2 | 75.9 | 33.0 | **60.8** |
| | | 20 | 55.0 | 77.5 | 41.5 | 74.0 | 68.0 | 74.9 | 32.2 | 60.4 |
| | | 40 | 54.3 | 77.3 | 41.1 | 73.3 | 66.5 | 73.7 | 31.8 | 59.7 |
| | | 80 | 52.6 | 76.8 | 39.5 | 71.7 | 65.4 | 71.0 | 31.8 | 58.4 |
| LLaMA-2 7B | 4 | 0 | 57.5 | 78.1 | 46.9 | 75.8 | 69.8 | 76.0 | 35.0 | 62.7 |
| | | 10 | 57.1 | 78.2 | 46.0 | 76.0 | 69.5 | 75.8 | 35.8 | 62.6 |
| | | **20** | 57.2 | 78.5 | 46.1 | 76.5 | 69.9 | 77.7 | 34.2 | **62.8** |
| | | 40 | 57.1 | 78.2 | 45.7 | 76.5 | 69.5 | 77.3 | 34.8 | 62.7 |
| | | 80 | 56.5 | 78.4 | 46.2 | 76.4 | 70.7 | 74.5 | 34.0 | 62.4 |
| | 3 | 0 | 55.6 | 77.9 | 42.1 | 74.3 | 69.5 | 72.0 | 34.0 | 60.8 |
| | | **10** | 55.7 | 78.6 | 44.0 | 74.2 | 68.4 | 72.5 | 34.8 | **61.2** |
| | | 20 | 55.5 | 77.8 | 43.0 | 74.6 | 69.0 | 71.7 | 34.2 | 60.8 |
| | | 40 | 55.4 | 77.8 | 43.4 | 73.6 | 68.9 | 71.7 | 35.0 | 60.8 |
| | | 80 | 53.9 | 76.8 | 42.1 | 73.7 | 67.3 | 71.8 | 34.4 | 60.0 |

Models with fewer than 20 LoRA warm-up steps performed best, as initializing LoRA parameters with minimal disturbance from quantization errors facilitated better convergence. Extending the LoRA warm-up beyond the quantization-aware fine-tuning phase, however, resulted in performance degradation, likely due to an insufficient number of training steps to fully compensate for quantization errors. Considering the variability in impact across different models and configurations, we generally applied 10 LoRA warm-up steps in our experiments.

# D   ABLATIVE STUDY ON LoRA RANK

We investigated the effect of LoRA rank size on L4Q training and found that a rank size of around 4 is sufficient for effective training. Using the LLaMA-1 7B model, we conducted training over 12,800 iterations with 128 batches. The remaining training conditions are consistent with the main experiments. The evaluation results for Commonsense QA and MMLU are presented in Table 6 and Table 7, respectively.

Table 6: Commonsense QA benchmark result on LLaMA-1 7B model. The numbers represent accuracy (%) for each task.

| Model | Rank | HellaSwag | PIQA | ARC-c | ARC-e | Winogrande | BoolQ | OBQA | Average |
|-------|------|-----------|------|-------|-------|------------|-------|------|---------|
| LLaMA-1 7B | 1 | 56.9 | 79.1 | 43.7 | 76.0 | 69.9 | 76.6 | 35.4 | 62.5 |
| | 2 | 58.6 | 78.6 | 45.4 | 76.8 | 69.9 | 74.1 | 36.4 | 62.8 |
| | 4 | 57.8 | 79.1 | 45.3 | 76.0 | 69.5 | 76.1 | 34.8 | 62.7 |
| | 8 | 58.5 | 79.1 | 45.8 | 76.5 | 70.2 | 75.8 | 37.0 | 62.7 |
| | 16 | 58.3 | 78.8 | 44.5 | 75.8 | 69.8 | 78.3 | 35.2 | 63.0 |
| | **32** | 58.6 | 79.0 | 45.2 | 76.3 | 70.2 | 76.7 | 37.6 | **63.4** |
| | 64 | 58.4 | 79.0 | 45.5 | 76.5 | 70.0 | 78.7 | 37.8 | 63.2 |
| | 128 | 58.6 | 79.1 | 45.9 | 76.7 | 70.6 | 75.8 | 36.4 | 63.3 |

Table 7: MMLU benchmark result on LLaMA-1 7B model. The numbers represent accuracy (%) for each category.

| Model | Rank | 0-shot | | | | | 5-shot | | | | |
|-------|------|--------|------|--------|--------|------|--------|------|--------|--------|------|
| | | Hums. | STEM | Social | Others | Avg. | Hums. | STEM | Social | Others | Avg. |
| LLaMA-1 7B | 1 | 31.8 | 29.2 | 32.9 | 37.0 | 32.7 | 32.3 | 30.7 | 35.2 | 39.9 | 34.4 |
| | 2 | 30.7 | 28.3 | 32.7 | 37.5 | 32.2 | 33.0 | 29.3 | 34.3 | 37.2 | 33.5 |
| | 4 | 32.4 | 32.1 | 36.7 | 39.4 | 34.9 | 32.9 | 31.4 | 38.7 | 39.5 | 35.4 |
| | **8** | 33.5 | 31.9 | 37.7 | 39.4 | 35.5 | 34.2 | 30.7 | 38.4 | 39.8 | **35.7** |
| | 16 | 30.0 | 31.6 | 35.1 | 38.1 | 33.5 | 32.2 | 31.2 | 35.3 | 38.1 | 34.8 |
| | 32 | 32.1 | 30.8 | 33.2 | 36.5 | 33.1 | 32.5 | 31.7 | 37.2 | 39.5 | 35.0 |
| | 64 | 33.4 | 31.2 | 36.1 | 38.8 | 34.8 | 33.6 | 31.1 | 35.9 | 39.5 | 35.0 |
| | 128 | 31.8 | 29.7 | 34.5 | 37.6 | 33.3 | 33.3 | 31.1 | 35.7 | 38.3 | 34.5 |

Increasing the rank beyond 32 or 4 does not lead to further performance improvements, which aligns with the observations in the original LoRA paper (Hu et al., 2022). Therefore, we generally applied a rank size of 4, considering that higher rank sizes introduce memory and computational overhead during training.

# E    EXPERIMENTAL SETTINGS

The baselines and L4Q are trained with AdamW optimizer Loshchilov & Hutter (2019) with a weight decay of 0.01. For the learning rate scheduler, a cosine decay scheduler with a linear warm-up through 10% of the total training steps. Learning rates are presented in Table 8.

Table 8: Learning rate conditions used to fine-tuning on each models for L4Q and baselines: QLoRA*, QA-LoRA, and QAT-LoRA.

| Model | Methods | | | |
| | QLoRA* | QA-LoRA | QAT-LoRA | L4Q |
|---|---|---|---|---|
| OpenLLaMA 3B | $1 \times 10^{-5}$ | $2 \times 10^{-5}$ | $5 \times 10^{-5}$ | $5 \times 10^{-5}$ |
| LLaMA-1 7B | $1 \times 10^{-5}$ | $2 \times 10^{-5}$ | $5 \times 10^{-5}$ | $5 \times 10^{-5}$ |
| LLaMA-1 13B | $1 \times 10^{-5}$ | $5 \times 10^{-5}$ | $4 \times 10^{-5}$ | $4 \times 10^{-5}$ |
| LLaMA-1 33B | $1 \times 10^{-5}$ | $5 \times 10^{-5}$ | $2 \times 10^{-4}$ | $2 \times 10^{-4}$ |
| LLaMA-2 7B | $2 \times 10^{-5}$ | - | $2 \times 10^{-4}$ | $2 \times 10^{-4}$ |
| LLaMA-2 13B | $2 \times 10^{-5}$ | - | $2 \times 10^{-4}$ | $2 \times 10^{-4}$ |

Batch size is set to 128. 50K iterations for the baselines that utilize PTQ applied schemes, such as QLoRA* and QA-LoRA, and 25K iterations for QAT involving schemes, such as QAT, QAT-LoRA, and L4Q are utilized to match the training latency. The training sequence length of training is set to above the maximum sequence length of the dataset, which is 2048, except for a 33B model with L4Q that is set to 128.

# F    TRAINING TIME

We report the total training time of L4Q and the quantization-aware PEFT baselines, QLoRA* and QA-LoRA, in Table 9. L4Q demonstrates similar time performance to the baselines, highlighting its scalability and time efficiency for larger model sizes, comparable to that of the baseline methods.

Table 9: Training time (in hours) spent on fine-tuning on OpenLLaMA and LLaMA-1 models with a A100 GPU.

| Methods | OpenLLaMA | | LLaMA-1 | |
| | 3B | 7B | 13B | 33B |
|---|---|---|---|---|
| QLoRA* | 4.5 | 9.9 | 18.0 | 38.4 |
| QA-LoRA | 5.0 | 11.2 | 19.8 | 36.2 |
| L4Q | 4.4 | 10.1 | 16.9 | 39.6 |

## G    THROUGHPUT AND SPEEDUP OF FULLY-QUANTIZED MODELS AND MIXED-PRECISION MODELS

We investigate the throughput and speedup of fully-quantized models and mixed-precision models, demonstrating that, although the number of LoRA parameters is negligible, it causes a noticeable drop in throughput when its forward path is not merged with that of the base linear layer. Using the LLaMA-1 7B, 13B, and 33B models, we conducted experiments to measure throughput (tokens per second) and compare speedup. In both fully-quantized and mixed-precision models, uniform quantization is applied to the linear layers, except for the head layer (lm_head), using the EXLLaMA2 kernel[4], which is designed for 4-bit weight-only quantized inference. For fp16 computations in LoRA within mixed-precision models or the baseline, default GEMM kernels are used. We measured the elapsed time for inferencing 512 tokens over 2000 data points with batch sizes ranging from 1 to 64, calculating throughput by dividing the number of tokens, which is set to be 512, by the elapsed time. The results are presented in Table 10.

Table 10: Throughput (tokens/sec) and Speedup for LLaMA models.  L4Q represents fully-quantized models, and QLoRA* represents mixed-precision models.  'OOM' indicates out-of-memory cases.

| Model | Method | Batch size | | | | | | | Speedup |
| | | 1 | 2 | 4 | 8 | 16 | 32 | 64 | |
|---|---|---|---|---|---|---|---|---|---|
| LLaMA-1 7B | **L4Q** | 38.04 | 75.12 | 148.63 | 216.51 | 255.64 | 276.43 | 318.43 | **1.81** |
| | QLoRA* | 24.80 | 47.88 | 96.51 | 184.06 | 234.79 | 247.79 | 299.94 | 1.33 |
| | None | 17.04 | 33.81 | 67.27 | 124.13 | 199.19 | 241.31 | OOM | 1.00 |
| LLaMA-1 13B | **L4Q** | 30.68 | 59.53 | 115.78 | 144.44 | 160.95 | 191.20 | OOM | **1.92** |
| | QLoRA* | 19.71 | 38.90 | 77.83 | 128.67 | 150.72 | 156.16 | OOM | 1.41 |
| | None | 13.67 | 26.97 | 53.23 | 85.47 | 124.17 | OOM | OOM | 1.00 |
| LLaMA-1 30B | **L4Q** | 20.43 | 40.05 | 64.00 | 73.29 | 79.77 | OOM | OOM | **2.25** |
| | QLoRA* | 13.22 | 25.48 | 50.81 | 66.89 | 75.11 | OOM | OOM | 1.44 |
| | None | 9.13 | 17.68 | OOM | OOM | OOM | OOM | OOM | 1.00 |

Fully-quantized models demonstrate a speedup of over 1.8x, while mixed-precision models achieve a maximum speedup of 1.4x, despite using the same quantization scheme and execution kernel, compared to the fp16 baselines. As a result, fully-quantized models achieve a 30% to 50% greater speedup compared to mixed-precision models. This demonstrates that L4Q, which produces fully-quantized models, offers higher inference efficiency and better hardware utilization than conventional quantization-aware PEFT methods, such as QLoRA and LoftQ, which retain unmerged forward paths for LoRA.

## H    DETAILED RESULT ON MAIN EVALUATIONS

We present the Commonsense QA and MMLU benchmark results with averaged accuracy score on Section 4. We present the detailed results of each benchmarks composed of several categories of tasks in Table 11 and Table 12 below. Through evaluation, we demonstrate that L4Q generally achieves higher accuracy in low-bit quantized models compared to both PTQ methods and PTQ-based fine-tuning methods. Notably, L4Q surpasses the pre-trained models on the Commonsense QA benchmarks and on the MMLU benchmarks with LLaMA-1 7B and 33B models. In contrast, PTQ-based fine-tuning methods, including those that incorporate high-precision LoRA weights, show lower performance compared to both L4Q and the pre-trained models. These results emphasize the challenges of recovering from quantization errors with PTQ alone and highlight the effectiveness of L4Q's joint quantization and fine-tuning scheme.

---

[4]https://github.com/turboderp/exllamav2.git

Table 11: Commonsense QA benchmark result. The numbers represent accuracy (%) of each task.

| Model | Method | #Bits | Hella. | PIQA | ARC-c | ARC-e | Winogr. | BoolQ | OBQA | Avg. |
|---|---|---|---|---|---|---|---|---|---|---|
| OpenLLaMA 3B | None | 16 | 48.8 | 75.0 | 33.9 | 69.2 | 61.6 | 66.9 | 28.2 | 54.8 |
| | LoRA | 16 | 49.8 | 75.6 | 37.0 | 70.2 | 63.1 | 68.0 | 27.2 | 55.9 |
| | GPTQ | 4 | 47.9 | 75.1 | 31.0 | 58.8 | 60.5 | 57.9 | 23.6 | 50.7 |
| | OmniQ | 4 | 48.2 | 73.8 | 33.1 | 69.5 | 60.1 | 67.5 | 26.6 | 54.1 |
| | LoftQ* | 4+16 | 48.0 | 74.5 | 34.0 | 68.6 | 60.9 | 67.2 | 26.0 | 54.2 |
| | QLoRA* | 4+16 | 48.4 | 74.3 | 33.0 | 69.4 | 61.5 | 67.1 | 26.8 | 54.4 |
| | QA-LoRA | 4 | 48.8 | 74.9 | 33.8 | 69.2 | 61.9 | 66.7 | 26.2 | 54.5 |
| | QAT-LoRA | 4+16 | 48.8 | 74.5 | 35.0 | 70.1 | 61.9 | 65.2 | 27.0 | 54.6 |
| | L4Q | 4 | 49.1 | 74.9 | 35.2 | 69.8 | 61.1 | 67.7 | 27.4 | 55.0 |
| | GPTQ | 3 | 46.3 | 72.6 | 31.8 | 64.7 | 58.1 | 66.5 | 25.6 | 52.2 |
| | OmniQ | 3 | 46.5 | 74.4 | 30.5 | 56.6 | 59.0 | 59.8 | 23.0 | 50.0 |
| | LoftQ* | 3+16 | 27.9 | 57.3 | 19.5 | 37.3 | 51.0 | 61.9 | 12.0 | 38.1 |
| | QLoRA* | 3+16 | 45.6 | 72.6 | 29.3 | 61.6 | 59.7 | 64.2 | 24.4 | 51.0 |
| | QA-LoRA | 3 | 46.3 | 72.6 | 28.9 | 66.0 | 59.5 | 63.4 | 23.8 | 51.5 |
| | QAT-LoRA | 3+16 | 46.7 | 74.1 | 33.2 | 67.2 | 60.5 | 64.1 | 26.4 | 53.2 |
| | L4Q | 3 | 47.2 | 75.0 | 32.3 | 68.3 | 60.9 | 67.2 | 27.0 | 54.0 |
| LLaMA 7B | None | 16 | 57.0 | 78.7 | 41.9 | 75.3 | 69.9 | 75.1 | 34.4 | 61.7 |
| | LoRA | 16 | 58.3 | 78.8 | 45.7 | 76.1 | 70.6 | 78.7 | 35.4 | 63.4 |
| | GPTQ | 4 | 53.9 | 77.7 | 40.3 | 73.5 | 67.9 | 72.9 | 30.0 | 59.4 |
| | OmniQ | 4 | 55.7 | 77.7 | 38.8 | 67.5 | 65.3 | 72.5 | 29.2 | 58.1 |
| | LoftQ* | 4+16 | 57.8 | 79.2 | 43.1 | 76.9 | 69.8 | 75.8 | 35.4 | 62.6 |
| | QLoRA* | 4+16 | 56.7 | 78.9 | 41.8 | 75.2 | 70.0 | 74.6 | 32.2 | 61.3 |
| | QA-LoRA | 4 | 57.2 | 78.9 | 41.2 | 74.9 | 70.6 | 73.6 | 32.6 | 61.3 |
| | QAT-LoRA | 4+16 | 57.7 | 78.9 | 44.7 | 75.3 | 68.9 | 75.8 | 35.6 | 62.4 |
| | L4Q | 4 | 57.8 | 79.1 | 45.3 | 76.0 | 69.5 | 76.1 | 34.8 | 62.7 |
| | GPTQ | 3 | 46.6 | 71.9 | 32.4 | 65.4 | 65.0 | 68.0 | 24.6 | 53.4 |
| | OmniQ | 3 | 54.0 | 77.1 | 35.6 | 64.9 | 64.7 | 71.2 | 28.0 | 56.5 |
| | LoftQ* | 3+16 | 43.4 | 68.9 | 33.0 | 65.5 | 56.5 | 58.5 | 23.0 | 49.8 |
| | QLoRA* | 3+16 | 53.9 | 76.2 | 39.3 | 71.5 | 68.9 | 72.8 | 31.0 | 59.1 |
| | QA-LoRA | 3 | 55.4 | 76.3 | 39.8 | 72.5 | 69.5 | 67.1 | 30.6 | 58.7 |
| | QAT-LoRA | 3+16 | 56.1 | 77.4 | 41.6 | 72.8 | 68.0 | 76.0 | 33.0 | 60.7 |
| | L4Q | 3 | 55.9 | 77.6 | 42.1 | 74.1 | 68.9 | 76.8 | 33.4 | 61.2 |
| LLaMA 13B | None | 16 | 59.9 | 79.2 | 46.5 | 77.4 | 72.8 | 78.0 | 33.2 | 63.8 |
| | LoRA | 16 | 60.8 | 79.7 | 50.3 | 78.6 | 72.3 | 80.2 | 34.8 | 65.2 |
| | GPTQ | 4 | 58.9 | 79.3 | 46.5 | 77.0 | 72.7 | 76.5 | 33.8 | 63.5 |
| | OmniQ | 4 | 58.6 | 79.7 | 43.8 | 73.5 | 70.5 | 68.7 | 28.4 | 60.4 |
| | LoftQ* | 4+16 | 60.6 | 79.0 | 48.3 | 77.7 | 72.9 | 76.0 | 35.0 | 64.2 |
| | QLoRA* | 4+16 | 59.6 | 79.2 | 46.5 | 77.1 | 72.5 | 78.1 | 33.4 | 63.8 |
| | QA-LoRA | 4 | 60.1 | 79.0 | 46.8 | 77.0 | 71.4 | 67.1 | 36.2 | 62.5 |
| | QAT-LoRA | 4+16 | 60.9 | 79.2 | 48.2 | 78.6 | 71.5 | 77.0 | 35.6 | 64.4 |
| | L4Q | 4 | 60.9 | 79.8 | 48.2 | 78.5 | 71.7 | 76.7 | 35.4 | 64.5 |
| | GPTQ | 3 | 57.3 | 77.3 | 42.6 | 73.0 | 71.0 | 74.6 | 31.4 | 61.0 |
| | OmniQ | 3 | 56.8 | 77.2 | 39.9 | 72.7 | 68.5 | 67.0 | 29.8 | 58.9 |
| | LoftQ* | 3+16 | 47.8 | 72.1 | 37.6 | 70.8 | 58.3 | 65.5 | 25.8 | 54.0 |
| | QLoRA* | 3+16 | 56.6 | 77.8 | 43.9 | 75.1 | 70.8 | 73.5 | 31.6 | 61.3 |
| | QA-LoRA | 3 | 57.7 | 78.0 | 44.7 | 75.3 | 71.2 | 68.6 | 32.4 | 61.1 |
| | QAT-LoRA | 3+16 | 59.1 | 78.1 | 46.3 | 77.0 | 70.8 | 74.7 | 36.2 | 63.2 |
| | L4Q | 3 | 58.9 | 78.4 | 45.8 | 77.4 | 70.2 | 77.7 | 35.2 | 63.4 |
| LLaMA 33B | None | 16 | 63.3 | 81.0 | 52.8 | 80.4 | 75.9 | 82.6 | 36.0 | 67.4 |
| | LoRA | 16 | 64.1 | 81.3 | 53.7 | 81.6 | 75.5 | 84.0 | 37.6 | 68.3 |
| | GPTQ | 4 | 61.8 | 80.5 | 49.1 | 78.9 | 73.6 | 82.2 | 33.6 | 65.7 |
| | OmniQ | 4 | 62.3 | 80.0 | 48.5 | 75.8 | 73.9 | 69.1 | 31.0 | 62.9 |
| | LoftQ* | 4+16 | 63.3 | 80.3 | 51.8 | 81.4 | 75.3 | 82.9 | 37.0 | 67.4 |
| | QLoRA* | 4+16 | 62.3 | 80.2 | 50.2 | 79.5 | 74.9 | 81.0 | 35.4 | 66.2 |
| | QA-LoRA | 4 | 62.8 | 80.3 | 50.1 | 79.5 | 75.1 | 73.2 | 36.4 | 65.3 |
| | QAT-LoRA | 4+16 | 62.3 | 81.3 | 53.0 | 81.3 | 74.9 | 82.7 | 35.4 | 67.3 |
| | L4Q | 4 | 63.9 | 81.0 | 53.0 | 81.3 | 75.0 | 82.8 | 35.8 | 67.5 |
| | LoftQ* | 3+16 | 46.1 | 74.9 | 38.9 | 73.9 | 58.3 | 63.5 | 28.2 | 54.8 |
| | QLoRA* | 3+16 | 59.6 | 78.7 | 46.0 | 76.8 | 72.5 | 81.6 | 34.6 | 64.3 |
| | QA-LoRA | 3 | 61.1 | 79.6 | 47.8 | 78.0 | 73.8 | 79.3 | 33.0 | 64.6 |
| | QAT-LoRA | 3+16 | 63.3 | 81.0 | 52.8 | 81.3 | 75.5 | 82.8 | 35.4 | 67.4 |
| | L4Q | 3 | 63.0 | 81.0 | 52.6 | 81.4 | 75.5 | 82.8 | 35.4 | 67.4 |
| LLaMA 2 7B | None | 16 | 57.1 | 78.1 | 43.4 | 76.3 | 69.1 | 77.7 | 31.4 | 61.9 |
| | LoRA | 16 | 57.9 | 78.9 | 48.0 | 77.4 | 70.3 | 75.8 | 34.8 | 63.3 |
| | GPTQ | 4 | 56.0 | 77.5 | 42.2 | 75.0 | 68.2 | 76.4 | 29.8 | 60.7 |
| | OmniQ | 4 | 56.0 | 77.7 | 41.3 | 69.9 | 67.8 | 73.5 | 30.2 | 59.5 |
| | LoftQ* | 4+16 | 57.0 | 78.0 | 43.3 | 76.3 | 69.2 | 76.8 | 31.4 | 61.7 |
| | QLoRA* | 4+16 | 56.6 | 77.8 | 43.3 | 75.2 | 69.1 | 75.3 | 31.8 | 61.3 |
| | QA-LoRA | 4 | 56.4 | 79.3 | 73.3 | 39.2 | 71.8 | 75.5 | 31.4 | 61.0 |
| | QAT-LoRA | 4+16 | 56.6 | 77.7 | 43.7 | 75.6 | 69.5 | 77.7 | 32.6 | 61.9 |
| | L4Q | 4 | 57.2 | 78.8 | 47.1 | 76.9 | 70.2 | 80.4 | 34.8 | 63.6 |
| | GPTQ | 3 | 53.1 | 76.2 | 35.8 | 70.3 | 67.7 | 72.4 | 27.6 | 57.6 |
| | OmniQ | 3 | 54.6 | 76.4 | 37.5 | 67.6 | 66.1 | 71.9 | 31.0 | 57.9 |
| | LoftQ* | 3+16 | 27.1 | 55.7 | 19.0 | 31.1 | 48.8 | 48.1 | 12.8 | 34.7 |
| | QLoRA* | 3+16 | 52.4 | 75.9 | 37.6 | 65.6 | 65.6 | 74.1 | 27.4 | 57.6 |
| | QA-LoRA | 3 | 56.5 | 77.8 | 42.3 | 74.7 | 68.0 | 30.8 | 43.8 | 56.3 |
| | QAT-LoRA | 3+16 | 52.0 | 75.2 | 39.3 | 71.1 | 65.1 | 69.9 | 29.3 | 57.4 |
| | L4Q | 3 | 55.5 | 77.3 | 42.8 | 73.8 | 68.8 | 77.2 | 34.0 | 61.3 |
| LLaMA 2 13B | None | 16 | 60.1 | 79.1 | 48.5 | 79.4 | 72.2 | 80.6 | 35.2 | 65.0 |
| | LoRA | 16 | 61.2 | 79.4 | 53.0 | 79.8 | 73.2 | 81.4 | 37.4 | 66.5 |
| | GPTQ | 4 | 59.5 | 78.3 | 47.3 | 78.7 | 72.1 | 80.9 | 34.2 | 64.4 |
| | OmniQ | 4 | 59.0 | 78.1 | 43.7 | 71.3 | 68.7 | 66.6 | 32.0 | 59.9 |
| | LoftQ* | 4+16 | 60.0 | 79.3 | 48.1 | 79.7 | 71.9 | 80.7 | 34.8 | 64.9 |
| | QLoRA* | 4+16 | 59.6 | 78.4 | 46.6 | 77.9 | 72.2 | 79.2 | 33.8 | 64.0 |
| | QA-LoRA | 4 | 59.4 | 78.5 | 79.1 | 46.9 | 72.3 | 80.7 | 34.4 | 64.5 |
| | QAT-LoRA | 4+16 | 59.5 | 78.8 | 48.4 | 79.2 | 71.5 | 80.9 | 34.4 | 64.7 |
| | L4Q | 4 | 60.9 | 80.1 | 51.2 | 79.7 | 71.0 | 82.2 | 35.8 | 65.8 |
| | GPTQ | 3 | 57.3 | 77.2 | 43.5 | 76.1 | 69.9 | 74.0 | 34.0 | 61.7 |
| | OmniQ | 3 | 57.8 | 78.2 | 42.0 | 72.3 | 68.0 | 69.9 | 31.2 | 59.9 |
| | LoftQ* | 3+16 | 28.7 | 60.6 | 19.5 | 45.3 | 50.7 | 55.1 | 15.2 | 39.3 |
| | QLoRA* | 3+16 | 57.8 | 77.9 | 44.3 | 76.7 | 70.0 | 78.1 | 32.6 | 62.5 |
| | QA-LoRA | 3 | 57.3 | 77.2 | 76.0 | 43.4 | 70.1 | 73.7 | 34.0 | 61.7 |
| | QAT-LoRA | 3+16 | 55.8 | 77.1 | 67.6 | 76.0 | 67.6 | 75.1 | 30.8 | 64.3 |
| | L4Q | 3 | 59.3 | 78.7 | 51.2 | 78.5 | 70.6 | 79.9 | 37.4 | 65.1 |

Table 12: MMLU benchmark result. The numbers represent accuracy(%) of each task.

| Model | Method | #Bits | 0-shot | | | | | 5-shot | | | | |
|---|---|---|---|---|---|---|---|---|---|---|---|---|
| | | | Human. | STEM | Social | Others | Avg. | Human. | STEM | Social | Others | Avg. |
| LLaMA 7B | None | 16 | 32.9 | 26.9 | 32.1 | 37.3 | 32.5 | 33.9 | 30.6 | 38.2 | 38.2 | 35.1 |
| | LoRA | 16 | 36.1 | 31.5 | 36.9 | 40.6 | 36.3 | 34.4 | 30.3 | 39.9 | 43.1 | 36.7 |
| | GPTQ | 4 | 28.4 | 27.1 | 27.0 | 30.4 | 28.3 | 31.5 | 30.4 | 33.7 | 35.7 | 32.7 |
| | OmniQ | 4 | 31.1 | 26.7 | 29.8 | 35.5 | 30.9 | 31.1 | 29.8 | 35.5 | 37.5 | 33.3 |
| | LoftQ* | 4+16 | 32.3 | 30.0 | 32.7 | 37.3 | 33.0 | 33.6 | 30.7 | 37.2 | 39.0 | 35.1 |
| | QLoRA* | 4+16 | 33.1 | 27.1 | 33.1 | 37.5 | 32.8 | 32.3 | 29.0 | 35.4 | 38.0 | 33.6 |
| | QA-LoRA | 4+16 | 33.5 | 29.5 | 34.6 | 37.4 | 33.8 | 32.2 | 32.4 | 35.6 | 39.9 | 34.8 |
| | QAT-LoRA | 4 | 33.5 | 29.5 | 37.5 | 37.9 | 34.5 | 34.1 | 31.2 | 38.5 | 39.0 | 35.6 |
| | L4Q | 4 | 32.4 | 32.1 | 36.7 | 39.4 | 34.9 | 34.2 | 30.7 | 38.4 | 39.8 | 35.7 |
| | GPTQ | 3 | 25.0 | 22.5 | 22.0 | 24.5 | 23.7 | 25.9 | 25.7 | 28.2 | 29.7 | 27.3 |
| | OmniQ | 3 | 27.8 | 29.7 | 26.8 | 32.2 | 29.0 | 31.6 | 32.1 | 33.7 | 29.7 | 31.6 |
| | LoftQ* | 3+16 | 24.3 | 21.7 | 22.4 | 24.5 | 23.4 | 23.4 | 23.2 | 21.9 | 23.5 | 23.1 |
| | QLoRA* | 3+16 | 27.8 | 27.1 | 26.6 | 29.1 | 27.7 | 30.5 | 28.6 | 32.1 | 34.9 | 31.5 |
| | QA-LoRA | 3+16 | 28.2 | 29.7 | 32.0 | 33.7 | 30.6 | 29.8 | 29.2 | 32.2 | 34.6 | 31.5 |
| | QAT-LoRA | 3 | 28.9 | 27.1 | 25.8 | 29.6 | 28.0 | 29.1 | 26.6 | 29.7 | 31.1 | 29.1 |
| | L4Q | 3 | 29.5 | 27.8 | 32.1 | 33.3 | 30.6 | 29.3 | 31.0 | 33.5 | 30.4 | 31.8 |
| LLaMA 13B | None | 16 | 41.0 | 36.5 | 49.3 | 48.6 | 43.6 | 43.8 | 35.3 | 52.7 | 54.2 | 46.3 |
| | LoRA | 16 | 42.4 | 34.0 | 49.4 | 51.9 | 44.3 | 45.0 | 36.4 | 54.1 | 53.1 | 47.0 |
| | GPTQ | 4 | 33.5 | 34.5 | 44.9 | 44.9 | 40.1 | 43.1 | 35.9 | 52.8 | 51.9 | 45.7 |
| | OmniQ | 4 | 39.8 | 35.1 | 48.6 | 48.1 | 42.6 | 43.1 | 35.7 | 52.5 | 52.3 | 45.7 |
| | LoftQ* | 4+16 | 39.0 | 34.8 | 47.8 | 48.5 | 42.4 | 43.4 | 34.3 | 52.3 | 52.1 | 45.4 |
| | QLoRA* | 4+16 | 39.0 | 35.7 | 47.5 | 47.2 | 42.1 | 43.8 | 35.3 | 52.0 | 52.8 | 45.9 |
| | QA-LoRA | 4+16 | 39.8 | 33.9 | 46.9 | 48.3 | 42.0 | 43.2 | 35.0 | 51.8 | 52.8 | 45.5 |
| | QAT-LoRA | 4 | 35.3 | 35.2 | 47.7 | 47.9 | 42.4 | 43.0 | 34.6 | 53.0 | 53.6 | 45.8 |
| | L4Q | 4 | 40.5 | 35.3 | 49.5 | 48.5 | 43.2 | 43.4 | 36.1 | 52.3 | 53.2 | 46.0 |
| | GPTQ | 3 | 32.1 | 27.2 | 36.3 | 36.8 | 33.1 | 36.0 | 29.8 | 42.2 | 45.6 | 38.2 |
| | OmniQ | 3 | 32.6 | 28.2 | 37.1 | 41.8 | 34.8 | 39.1 | 33.1 | 47.5 | 47.6 | 41.6 |
| | LoftQ* | 3+16 | 25.1 | 24.7 | 24.0 | 26.1 | 25.0 | 24.9 | 27.4 | 24.1 | 25.0 | 25.3 |
| | QLoRA* | 3+16 | 33.3 | 31.3 | 38.5 | 42.3 | 36.1 | 36.8 | 32.4 | 46.6 | 47.0 | 40.4 |
| | QA-LoRA | 3+16 | 37.6 | 31.6 | 41.9 | 44.3 | 38.8 | 38.0 | 34.1 | 44.5 | 47.9 | 40.9 |
| | QAT-LoRA | 3 | 35.9 | 28.4 | 42.4 | 43.5 | 37.5 | 34.8 | 31.5 | 43.0 | 44.8 | 38.2 |
| | L4Q | 3 | 38.5 | 33.2 | 44.7 | 47.2 | 40.7 | 39.3 | 34.0 | 46.6 | 48.4 | 41.8 |
| LLaMA 33B | None | 16 | 51.0 | 40.1 | 62.2 | 59.4 | 53.0 | 54.4 | 44.7 | 65.4 | 61.6 | 56.4 |
| | LoRA | 16 | 49.2 | 41.3 | 61.4 | 58.7 | 54.4 | 55.2 | 46.1 | 66.4 | 63.3 | 57.6 |
| | GPTQ | 4 | 49.4 | 39.6 | 59.1 | 58.1 | 51.4 | 52.5 | 45.1 | 64.2 | 62.2 | 55.7 |
| | OmniQ | 4 | 48.5 | 40.3 | 61.3 | 59.1 | 52.0 | 53.4 | 44.8 | 64.7 | 61.1 | 55.4 |
| | LoftQ* | 4+16 | 49.2 | 40.2 | 60.8 | 58.0 | 51.8 | 54.6 | 44.5 | 65.2 | 61.5 | 56.4 |
| | QLoRA* | 4+16 | 48.5 | 39.0 | 59.7 | 57.8 | 51.0 | 54.5 | 44.2 | 63.4 | 60.5 | 55.6 |
| | QA-LoRA | 4+16 | 50.2 | 39.8 | 60.9 | 58.9 | 52.3 | 55.0 | 45.5 | 65.1 | 61.8 | 56.7 |
| | QAT-LoRA | 4 | 45.2 | 39.7 | 56.6 | 55.5 | 48.9 | 52.7 | 43.5 | 63.4 | 61.0 | 55.0 |
| | L4Q | 4 | 50.8 | 42.1 | 61.5 | 59.4 | 53.3 | 53.5 | 46.6 | 66.1 | 61.8 | 56.7 |
| | LoftQ* | 3+16 | 24.7 | 24.0 | 23.2 | 26.4 | 24.6 | 24.3 | 23.2 | 22.9 | 25.6 | 24.0 |
| | QLoRA* | 3+16 | 41.8 | 34.6 | 55.2 | 52.3 | 45.6 | 46.4 | 40.9 | 57.9 | 56.7 | 50.1 |
| | QA-LoRA | 3+16 | 47.7 | 38.9 | 58.0 | 56.6 | 50.1 | 46.8 | 41.2 | 58.2 | 57.5 | 50.6 |
| | QAT-LoRA | 3 | 41.5 | 37.2 | 54.4 | 53.1 | 46.1 | 45.4 | 39.9 | 55.6 | 55.0 | 48.7 |
| | L4Q | 3 | 46.4 | 38.7 | 57.6 | 56.2 | 49.4 | 46.4 | 41.6 | 58.2 | 58.8 | 50.8 |
| LLaMA 2 7B | None | 16 | 39.3 | 34.0 | 47.9 | 46.0 | 41.6 | 42.8 | 37.0 | 50.6 | 52.2 | 45.4 |
| | LoRA | 16 | 41.0 | 34.6 | 50.8 | 45.9 | 43.9 | 43.4 | 37.0 | 51.8 | 52.4 | 46.0 |
| | GPTQ | 4 | 36.0 | 30.1 | 41.3 | 41.1 | 37.1 | 40.9 | 33.9 | 48.9 | 48.6 | 42.9 |
| | OmniQ | 4 | 37.7 | 34.6 | 47.2 | 45.7 | 41.0 | 42.4 | 37.7 | 51.1 | 51.6 | 45.4 |
| | LoftQ* | 4+16 | 36.7 | 31.3 | 42.9 | 43.6 | 38.5 | 41.5 | 34.6 | 49.5 | 49.8 | 43.7 |
| | QLoRA* | 4+16 | 37.3 | 31.5 | 43.3 | 42.7 | 38.6 | 42.1 | 35.9 | 50.2 | 51.1 | 44.6 |
| | QA-LoRA | 4+16 | 36.5 | 32.4 | 40.6 | 42.6 | 37.9 | 41.8 | 35.2 | 48.6 | 50.2 | 43.8 |
| | QAT-LoRA | 4 | 37.3 | 32.3 | 43.5 | 43.0 | 38.9 | 42.0 | 35.7 | 49.6 | 50.8 | 44.4 |
| | L4Q | 4 | 38.7 | 33.8 | 45.6 | 46.4 | 40.9 | 42.9 | 37.7 | 50.5 | 51.9 | 45.5 |
| | GPTQ | 3 | 28.9 | 28.5 | 35.3 | 33.7 | 31.3 | 36.0 | 31.7 | 39.3 | 43.5 | 37.5 |
| | OmniQ | 3 | 33.1 | 30.4 | 39.1 | 35.5 | 34.3 | 34.1 | 32.4 | 41.6 | 44.4 | 37.7 |
| | LoftQ* | 3+16 | 24.1 | 21.3 | 21.8 | 23.8 | 22.9 | 23.7 | 26.1 | 22.4 | 24.9 | 24.2 |
| | QLoRA* | 3+16 | 30.2 | 29.1 | 36.0 | 35.5 | 32.5 | 35.4 | 32.5 | 40.5 | 42.7 | 37.6 |
| | QA-LoRA | 3+16 | 31.1 | 27.2 | 33.9 | 33.8 | 31.5 | 34.2 | 31.2 | 39.9 | 42.7 | 36.8 |
| | QAT-LoRA | 3 | 28.9 | 27.8 | 34.7 | 33.7 | 31.0 | 36.0 | 31.6 | 39.5 | 43.4 | 37.5 |
| | L4Q | 3 | 31.0 | 32.7 | 38.6 | 39.2 | 34.9 | 34.3 | 32.3 | 42.2 | 44.9 | 38.0 |
| LLaMA 2 13B | None | 16 | 47.8 | 42.3 | 60.5 | 59.4 | 52.1 | 52.0 | 43.8 | 63.0 | 61.2 | 54.5 |
| | LoRA | 16 | 48.8 | 42.4 | 60.9 | 59.2 | 52.5 | 54.4 | 44.3 | 63.4 | 60.8 | 55.7 |
| | GPTQ | 4 | 46.5 | 40.2 | 57.7 | 56.8 | 50.0 | 52.3 | 43.1 | 62.7 | 61.5 | 54.7 |
| | OmniQ | 4 | 47.8 | 41.9 | 60.1 | 58.9 | 51.8 | 53.0 | 43.0 | 62.5 | 60.5 | 54.7 |
| | LoftQ* | 4+16 | 47.2 | 42.0 | 60.4 | 58.9 | 51.7 | 52.6 | 43.2 | 62.8 | 60.1 | 54.5 |
| | QLoRA* | 4+16 | 46.9 | 40.9 | 58.8 | 57.6 | 50.7 | 51.3 | 43.1 | 62.5 | 60.8 | 54.2 |
| | QA-LoRA | 4+16 | 47.5 | 41.0 | 58.8 | 56.8 | 50.7 | 50.3 | 42.9 | 62.3 | 60.7 | 53.8 |
| | QAT-LoRA | 4 | 46.5 | 40.8 | 58.3 | 57.4 | 50.4 | 51.6 | 42.5 | 62.3 | 60.7 | 54.1 |
| | L4Q | 4 | 48.4 | 41.8 | 60.4 | 58.4 | 51.9 | 53.6 | 44.3 | 62.7 | 60.5 | 55.2 |
| | GPTQ | 3 | 43.5 | 37.3 | 53.6 | 51.8 | 46.3 | 46.3 | 42.7 | 57.3 | 56.2 | 50.4 |
| | OmniQ | 3 | 42.3 | 38.9 | 54.5 | 51.3 | 46.3 | 43.4 | 43.0 | 58.8 | 56.5 | 50.2 |
| | LoftQ* | 3+16 | 24.2 | 21.7 | 23.8 | 23.7 | 23.5 | 24.6 | 28.5 | 24.0 | 27.4 | 26.0 |
| | QLoRA* | 3+16 | 43.9 | 38.3 | 53.9 | 52.2 | 46.8 | 48.5 | 41.1 | 57.7 | 55.9 | 50.6 |
| | QA-LoRA | 3+16 | 42.5 | 37.3 | 53.0 | 52.1 | 45.9 | 44.8 | 40.5 | 56.3 | 55.7 | 48.9 |
| | QAT-LoRA | 3 | 43.4 | 37.4 | 53.7 | 52.1 | 46.4 | 47.5 | 41.5 | 56.3 | 55.3 | 49.9 |
| | L4Q | 3 | 43.7 | 39.0 | 54.4 | 52.2 | 47.1 | 46.6 | 39.8 | 58.4 | 56.7 | 50.0 |

