# OpenReview forum: "L4Q: Parameter Efficient Quantization-Aware Fine-Tuning on Large Language Models"
_ICLR.cc/2025/Conference — ICLR 2025 Conference Withdrawn Submission_

### Official Review · Reviewer_ojJo · 2024-10-28

**Soundness:** 3
**Presentation:** 3
**Contribution:** 2
**Rating:** 5
**Confidence:** 4

**Summary:**

This paper introduces L4Q, a new variant of LoRA that effectively integrates Quantization-Aware Training (QAT) with LoRA. This approach minimizes quantization error during fine-tuning while maintaining the efficiency of quantized models in inference. Experimental results demonstrate its effectiveness.

**Strengths:**

1. Efficiently restoring the performance of quantized models is crucial.
2. The paper is well-written and easy to follow.
3. The proposed method is OK, and the experimental results are strong.

**Weaknesses:**

1. The method is incremental. I am not confident if the contributions are enough.
2. The proposed method seems sacrificed the training efficiency (training memory)  for a better performance
3. High similarity with existing method QLLM [1] or at least a special case.

[1] [1] QLLM: ACCURATE AND EFFICIENT LOW-BITWIDTH QUANTIZATION FOR LARGE LANGUAGE MODELS

**Questions:**

1. The proposed method appears to apply quantization after each training step. Has the author considered or compared the performance when applying quantization after a set number of steps?
2. The proposed method is somehow similar with the 4.2 section in QLLM[1], please provide some discussions and comparisons.
3. Did the authors employ strategies such as gradient checkpointing during gradient computation? If so, providing this information would allow for fairer comparisons.


[1] QLLM: ACCURATE AND EFFICIENT LOW-BITWIDTH QUANTIZATION FOR LARGE LANGUAGE MODELS

---

### Official Review · Reviewer_7QcK · 2024-11-01

**Soundness:** 1
**Presentation:** 1
**Contribution:** 2
**Rating:** 3
**Confidence:** 4

**Summary:**

This paper introduces L4Q, a quantization-aware fine-tuning framework that integrates QAT with LoRA to reduce quantization error in LLMs while achieving memory-efficient training. By adopting a memory-optimized layer design, L4Q significantly reduces the memory overhead of QAT while producing fully quantized weights, enabling efficient adaptation of downstream tasks. Experimental results show that L4Q achieves competitive accuracy and inference efficiency with fewer resource requirements in sub-4-bit quantization setups.

**Strengths:**

1. Combining QAT with LoRA is a great approach, as LoRA can be integrated with the quantized model without increasing inference time due to the retained LoRA branches.
2. The author provides specific implementation details and code to ensure the reproducibility of the method.

**Weaknesses:**

I believe the main weaknesses with this paper is that the experiments are insufficient to validate the effectiveness of the L4Q method, including but not limited to the following weaknesses:

1. The data in this paper does not match that of the original study. For example, LLaMA-1-7B with 4-bit quantization and group size is 128 achieves 38.4% accuracy on the MMLU (5-shot) benchmark using QA-LoRA (Table 6 in QA-LoRA, an improvement of 3.8% over the fp16 LLaMA-1-7B baseline), but in this paper, QA-LoRA’s accuracy is reported as only 35.6% (an improvement of 0.5% over the fp16 LLaMA-1-7B baseline). Additionally, L4Q achieves only 35.7% accuracy (an improvement of 0.6% over the fp16 LLaMA-1-7B baseline), making it noticeably less effective than QA-LoRA. This is just one example. The data in Tables 2 and 3 show significantly lower accuracy compared to the data in the QA-LoRA paper, including both the baseline methods and the L4Q method, and based on the original QA-LoRA data, L4Q's accuracy is significantly lower by comparison.
2. The comparison baseline is outdated and a more advanced baseline [1] should be included in the discussion. I found that L4Q performs significantly worse than IR-QLoRA [1]. For example, LLaMA-1-7B with 4-bit quantization achieves 40.8% accuracy on the MMLU (5-shot) benchmark using IR-QLoRA (an improvement of 6.2% over the fp16 LLaMA-1-7B baseline), whereas L4Q achieves only 35.7% accuracy (an improvement of just 0.6% over the fp16 LLaMA-1-7B baseline).
3. The authors have not demonstrated the benefits of quantization parameter initialization on final accuracy. They should show the accuracy of L4Q without quantization parameter initialization to verify its effectiveness.
4. Concerns about the effectiveness of LoRA warm-up in the L4Q method. From the data in Table 5, warm-up has basically no improvement on the average accuracy of L4Q on 7 common sense question answering benchmarks or even worse (-0.3%, 0%, 0.1% and 0.4%).
5. QA-LoRA shows 2-bit experimental data in their paper. Since the authors stated in the abstract “particularly in sub-4-bit quantization, positioning L4Q as an efficient QAT solution”, the authors should also show the experimental data of L4Q at 2-bit to verify the above statement.
6. The authors used different quantization group sizes for the LLaMA and OpenLLaMA models. Quantization group size is a hyperparameter that controls the number of parameters for quantization and low-rank adaptation. The authors should add ablation experiments to show the impact of quantization group size on the final accuracy.
7. The selected LLaMA 1 and 2 models are outdated. The authors should verify the effectiveness of the L4Q method on the more advanced LLaMA 3 model.
8. The authors only conducted experiments on the LLaMA model family and lacked verification of the effectiveness of the L4Q method on models with non-LLaMA architectures (such as the OPT model).
9. In Table 12, QA-LoRA is 4+16 bits, which is wrong and should be 4 bits.

Due to the above weaknesses, I have significant concerns regarding the effectiveness of the L4Q method, and therefore, I do not believe this paper is ready to be accepted. I would be glad to discuss my evaluation with the authors and look forward to their response.

[1] Accurate LoRA-Finetuning Quantization of LLMs via Information Retention. ICML 2024.

**Questions:**

I hope the authors can address the above weaknesses, and I also have the following questions for which I would appreciate their response.

1. The data in Table 9 show that the training time of L4Q is better than QA-LoRA in the case of 3-13B, but in the case of 33B, the training time of L4Q is 3.4 hours longer than QA-LoRA. I think that since the training cost of smaller models is less than QA-LoRA, the training cost of larger models should also be smaller. Could the authors explain why training costs are higher for larger models? In addition, QA-LoRA is PTQ-based LoRA parameter training, and L4Q is QAT-based LoRA & quantization parameter training. Logically, L4Q should take longer to train than QA-LoRA. Can the author explain my doubts?
2. How did the authors measure the speedup ratio of the quantized model? In Figure 4, the experimental data shows that under 4-bit quantization, both L4Q and QA-LoRA achieved the same speedup ratio. However, QA-LoRA uses min-max quantization (Equation 1 in the QA-LoRA paper), which differs from the quantization format used by the authors in Equation 3. Logically, the speedup ratios should not be exactly the same. Can the authors provide an explanation?

---

### Official Review · Reviewer_RARA · 2024-11-01

**Soundness:** 3
**Presentation:** 3
**Contribution:** 2
**Rating:** 5
**Confidence:** 4

**Summary:**

In this paper the authors introduce L4Q, a parameter-efficient quantization-aware fine-tuning technique for LLMs. For this they combine elements from PEFT literature such as Lora/QLora with elements from traditional QAT literature. The key benefit of this is that they minimize memory overhead during training and inference, while prior literature typically focused on only one of them. To do so, they integrate Lora inside the QAT formulation which effectively reduces the quantization noise. They show on various Llama models that L4Q is more accurate than existing PTQ techniques while being more inference efficient than PEFT techniques.

**Strengths:**

* Doing memory efficient QAT and considering inference efficiency for PEFT is very relevant and timely. Only limited literature addresses the combination of both.
* While the method is simple, it seems to be effective based on the authors evaluation.
* It is good that they show the actual inference speedup during inference, this is not always the case in the quantization literature.
* Paper is well written and easy to follow.

**Weaknesses:**

* The main weakness of this work is the limited technical novelty as L4Q can be seen as a straight forward integration of Lora and QAT (Lora adapter move insight round/clip of the quantization). That being said, I do acknowledge this can come with several engineering challenges to make it actually memory efficient in practice.
* For the results, some baselines and literature comparisons are missing, e.g. traditional QAT (LSQ, LSQ+), other LLM-based QAT work (e. g. LLM-QAT [1]) and more recent PTQ baseline such as SpinQuant [2]. While comparison to traditional QAT is significantly more memory expensive, it would be important to understand how much one might benefit when using the extra memory instead of relying on just fine-tuning low rank adapters.
* Minor:
    * Related work is not sufficiently discussed. While the background section the authors discusses several directly relevant works, a proper positioning of L4Q compared to other LLM quantization literature is missing (and also comparison to PEFT literature is limited).
    * The extensive discussions of the gradients in the method section, eq 9 - 11, seem fairly obvious and comes out straight from the chain rule. It is unclear to me what the authors want to say with them to the reader, especially since any deep learning framework can calculated them automatically (assuming STE is implemented).
    * The novel quantization initialization seems a bit of an over-claim, there is a plethora of initialization methods in literature and various quantization frameworks, and many varieties of min-max based once (also the standard symmetric baseline they compare to should be dividing by 2^(n-1) - 1 and not 2^(n-1) ).
    * The proposed method is very similar to the recent LR-QAT [3]. As this got only published ~4 months before the submission deadline, the methods might have been developed concurrently, thought a comparison should still be made.
    * Source code is not released/submitted which reduces the reproducibility and easy follow up research.


References:
* [1] LLM-QAT: Data-Free Quantization Aware Training for Large Language Models, Liu et al., 2023, https://arxiv.org/abs/2305.17888
* [2] SpinQuant: LLM quantization with learned rotations, Lui et al. 2024, https://www.arxiv.org/abs/2405.16406
* [3] Low-Rank Quantization-Aware Training for LLMs, Bondarenko et al. 2024, https://arxiv.org/abs/2406.06385

**Questions:**

* In the comparison to GPTQ and OmniQuant (tbl 2, 3) it is a bit unclear to me how the authors obtained the numbers stated. The common sense reasoning results seem to not align with the numbers stated in the OmniQuant paper, could you please elaborate?
* Similar to the above, are the results from LoftQ, QLora and QA-Lora from the corresponding papers or trained using the exact same pipeline used by L4Q?
* I do encourage the authors to extend the evaluation to more recent LLMs (e.g. Llama v3, Phi-3, Mistral)

---

### Official Review · Reviewer_PeCT · 2024-11-01

**Soundness:** 3
**Presentation:** 3
**Contribution:** 3
**Rating:** 5
**Confidence:** 4

**Summary:**

The paper aims to reduce the costs of LLMs by compressing them, specifically, discusses the quantization-aware training technique for reducing inference costs with parameter-efficient fine-tuning (PEFT) methods like Low-Rank Adaptation (LoRA) for reducing training costs. The focus is on improving quantization-aware PEFT approaches, which traditionally rely on a two-step process—post-training quantization (PTQ) followed by PEFT—that struggles with performance due to quantization errors and a lack of full quantization efficiency. The authors propose a new approach, L4Q, that combines Quantization-Aware Training (QAT) with LoRA to handle these issues. L4Q’s design optimizes memory use in QAT and achieves fully-quantized weights, making it highly effective for adapting models to new tasks while preserving accuracy. Tested on LLaMA families, L4Q shows accuracy improvements, especially in sub-4-bit quantization.

**Strengths:**

- It is crucial to improve the efficiency of quantization-aware training on LLMs, as it can bring more accurate yet low-precision models.
- The paper is well-written and easy to understand.

**Weaknesses:**

- The discussion of QAT-LoRA conceptually overlaps with some discussions in QA-LoRA, e.g., the precision mismatching of the LoRA module and dense layer. Moreover, simply inserting the LoRA module into the quantizer seems too straightforward, which further raises more undiscussed issues. Firstly, how are the LoRA modules inited? Since the LoRA modules are inside the quantizer, can we use quantization error to init the A and B, as in LoftQ? Secondly, why the training time costs are not increased compared with other PEFT methods that do not insert the LoRA modules into the quantizer? In my view, the gradient should pass the quantizer and thus the training costs should change. I indeed saw some time costs of training in the appendix but could the authors give a more detailed analysis of this?
- Some results are a bit confusing for me. For example, in the original paper, 4bit-llama33b-QALoRA results on MMLU 0-shot and 5-shot are 55.4 and 58.1, respectively, but in L4Q the results are only 48.9 and 55.0, respectively.
- Minor: The background part is too redundant. The derivatives of the scaling factor and bias are unnecessary, as they are uninformative to this paper.  Moreover, some notations should be carefully refined. For example, R is used to represent a function and A is used to represent a matrix, but they are both shown with capital letters.

**Questions:**

Please refer to the weakness.

---

### Note · Authors · 2024-11-21

**Comment:**

We sincerely appreciate the reviewers' thoughtful comments and valuable suggestions. We will improve our work and resubmit.

**Withdrawal Confirmation:**

I have read and agree with the venue's withdrawal policy on behalf of myself and my co-authors.